

# Climate Change Impacts on Yangtze River Discharge at the Three Gorges Dam

Steve J. Birkinshaw[1], Selma B. Guerreiro[1], Alex Nicholson[2], Qiuhua Liang[1], Paul Quinn[1], Lili Zhang[3], Bin He[4], Junxian Yin[3], Hayley J. Fowler[1]

5  [1]School of Civil Engineering and Geosciences, Newcastle University, Newcastle NE1 7RU, UK
[2]Ove Arup and Partners, Admiral House, 78 East St., Leeds, UK
[3]State Key Laboratory of Simulation and Regulation of Water Cycle in River Bain, China Institute of Water Resources and Hydropower Research, Beijing, 100038, China
[4]School of Hydraulic Engineering, Dalian University of Technology, Dalian, P.R. China

*Correspondence to*: Steve J. Birkinshaw (s.j.birkinshaw@ncl.ac.uk)

**Abstract.** The Yangtze River Basin is home to more than 400 million people, contributes to nearly half of China's food production, and is susceptible to major floods. Therefore planning for climate change impacts on river discharges is essential.  We used a physically-based distributed hydrological model, Shetran, to simulate discharge in the Yangtze River
just below the Three Gorges Dam at Yichang (1,007,200 km$^2$), obtaining  an excellent match between simulated and measured daily discharge, with Nash-Sutcliffe efficiencies of 0.95 for the calibration period (1996-2000) and 0.92 for the validation period (2001-2005). We then used a simple monthly delta change approach for 78 climate model projections (35 different GCMs) from the Coupled Model Intercomparison Project-5 (CMIP5) to examine the effect of climate change on river discharge for 2041-2070 for Representative Concentration Pathway 8.5. Projected changes to the basin's annual
precipitation varied between -3.6% and +14.8% but increases in temperature and consequently evapotranspiration (calculated using the Thornthwaite equation) were projected by all CMIP5 models, resulting in projected changes in the basin's annual discharge from -29.8% to +16.0%. These large differences were mainly due to the predicted expansion of the summer monsoon north and west into the Yangtze basin in some CMIP5 models, e.g. CanESM2, but not in others, e.g. CSIRO-Mk3-6-0. This was despite both models being able to simulate current climate well. Until projections of the strength
and location of the monsoon under a future climate improve there will remain large uncertainties in the direction and magnitude of future change in discharge for the Yangtze

## 1 Introduction

The Yangtze (or Chang Jiang) River (Fig. 1) is the third longest river in the world (6418 km) and the longest river in Eurasia.
Its source is located on the Qinghai-Tibet Plateau, at 5100m elevation, and extends to the East China Sea through the city of Shanghai. The River basin covers an area of 1,808,500 km$^2$, and is home to a population greater than 400 million (Dai et al.,



2012). Industry and agriculture within the Yangtze River basin generates 30-40% of China's GDP and the Yangtze River basin contributes nearly half of China's crop production, including more than two-thirds of the total volume of rice (Yang et al. 2005). The Yangtze River has been responsible for some of China's worst natural disasters. Catastrophic floods occurring over the last century include events in 1911, 1931, 1935 and 1954, which claimed the lives of over 300,000 people. As

recently as 1998, flooding of the Yangtze River caused over 4000 deaths, inundated 250,000 km$^2$ of agricultural land, and cost in excess of \$36 billion in damage to property and infrastructure (Yin and Li, 2001).

The Three Gorges Dam (TGD), which is located near Yichang, is the largest of more than 50,000 dams which have been built in the Yangtze basin (Li et al. 2013). The TGD reservoir is 600km long with a surface area of 1,084 km$^2$ and storage of 39.3 km$^3$ of water (Dai et al. 2006). It was built to help alleviate flooding on the Yangtze plain, for hydroelectric power

generation and to improve upstream navigation. Construction finished in 2012, when it was the largest hydropower dam in the world in terms of installed capacity, with a maximum output of 23,200MW from 34 turbines (Dai et al., 2006).

Over recent decades the Yangtze has been at a boundary between decreasing precipitation in the north-east of China and increasing precipitation in the south-east (Zhang et al. 2011a,b) due to the weakening of the East Asian summer monsoon (Wang et al. 2012). This has caused a small, but statistically insignificant, increase in discharge for the Yangtze basin since

1960 but a persistent decrease further north in the Yellow river basin (Piao et al. 2010). Therefore, it is instructive to examine how climate change may impact river discharges in the Yangtze basin, as any modifications to the seasonal distribution of precipitation and temperature may also have a major effect on flooding, water resources and hydro-power generation in the TGD.

Using the most recent CMIP5 climate change projections (Taylor et al. 2012) there have been a number of studies

considering future precipitation in the Yangtze. Tian et al. (2015) showed that for 22 CMIP5 models under RCP8.5 there was an overall increase in precipitation in China, but these increases were larger further north and only small increases were projected for the Yangtze. Similar results are also shown in Piao et al. (2010), Tao et al. (2012) and Jiang and Tian (2013). However, there have been no previous studies using the most recent CMIP5 climate change projections together with a hydrological model. Ma et al. (2010) considered terrestrial water storage changes within the Yangtze basin using the

Variable Infiltration Capacity (VIC) macroscale hydrological model under the SRES A2 and B2 (Nakićenović et al., 2000) climate scenarios. These scenarios showed that the south-east and central parts of the basin had the highest annual variations in storage. Koirala et al. (2014) considered runoff from 11 CMIP5 models together with a routing model. They found little change in discharge from the Yangtze basin but higher discharges further north in China, due to the increased projected precipitation.

In this paper, the Shetran physically-based distributed hydrological model is used to simulate river discharge for the Yangtze basin (Fig. 1) to the TGD near Yichang (1,007,200 km2) for ten years from 1/1/1996 to 31/12/2005. Other hydrological models have previously been applied to the Yangtze basin (Hayashi et al. 2008, Woo et al. 2009, Xu et al. 2008), but in terms of grid resolution, this is the most detailed hydrological model that has been produced for a major part of this basin. Shetran is then run under a changed climate using a simple monthly delta change approach on the outputs of 35 atmosphere-



ocean general circulation models (GCMs) (78 individual projections) from CMIP5 under Representative Concentration Pathway (RCP) 8.5. The results from the CanESM2 and CSIRO-Mk3-6-0 models are then considered in detail.

## 2 Data and Methods

### 2.1 Time Series Data

Most of the Yangtze (apart from the Tibetan plateau) has a sub-tropical monsoon climate. This has a distinct wet season (May – September) with high precipitation totals and high temperatures. We use observed daily data for 1/1/1996 – 31/12/2005 for 64 precipitation stations, 90 air temperature stations and 52 potential evapotranspiration stations (see Fig. 1 for station locations), with Thiessen polygons used to assign the spatial distribution in each case. Figure 2 shows the annual cycle of precipitation, potential evapotranspiration, measured and simulated discharge totals in the Yangtze basin to Yichang
from 1996-2005. The highest precipitation and potential evapotranspiration totals are in July; discharge totals are highest from July to September.

Figure 3a shows the Theissen polygon annual precipitation totals over the Yangtze basin. Annual totals vary from 370mm on the Tibetan plateau, up to 1400mm near the TGD. The highest annual precipitation, 1700mm, is observed at gauge 56385 at the western edge of the Sichuan Basin; at only 100km from the highest point in the basin at Mount Gongga (7556m), there
may be some orographic effects at this location. Figure 3b shows Theissen polygon mean annual air temperature over the Yangtze basin. Temperature shows considerable spatial variation across the basin with mean average annual air temperature ranging from $-4.5^0$C on the Tibetan plateau to $21.3^0$ C towards the southern edge of the basin. Average monthly temperature over the basin ranges from $-5^0$C in January to $16^0$C in July. Daily potential evapotranspiration also shows considerable spatial variation, ranging from an annual total of 600mm on the Tibetan plateau to 1300mm near the TGD.
We also used daily discharge data from Yichang for 1/1/1996 to 31/12/2005 (see Fig. 1 for location). Yichang is downstream of the TGD and in May 2003 the dam began to retain water. More details of the effect of the dam on discharges at Yichang are discussed in Sect. 3.1.

### 2.2 Shetran

SHETRAN (http://research.ncl.ac.uk/shetran/) is a physically-based distributed modelling system for water flow, sediment
and solute transport in river basins (Ewen et al., 2000; Birkinshaw et al., 2010). The most convenient way of visualizing SHETRAN is as a set of vertical columns with each column divided into finite-difference cells. There are 10,072 vertical columns, each of which is 10km by 10km, with each column divided into up to 25 finite-difference cells (making a total of around 250,000 finite difference cells). The lower cells contain aquifer materials and groundwater, higher cells contain soil and soil water and the uppermost cells contain surface waters and the vegetation canopy. River channels are specified around
the edge of the finite-difference columns and the location and elevations of these channels were calculated automatically using the method demonstrated in Birkinshaw (2010). Overall, 4143 river channel sections were specified.





Digital elevation model data was extracted from the STRM 90m grid resolution dataset (http://srtm.csi.cgiar.org/). Land-use for each grid square was obtained from the 1km resolution Global Land Cover map for the year 2000 (Bartholome et al., 2002), with the data acquired from an instrument on board the SPOT 4 satellite. The Asian dataset has 31 classes, although some of these were not present over the Yangtze basin and some were present in very small numbers. Overall there were seven main categories used in the Shetran simulations (Table 1). In the high elevation Tibetan plateau the main vegetation is shrub/herbaceous and deciduous forest. The rest of the basin is mostly cropland and rice paddies with evergreen forest around the steep edge of the Sichuan basin. Most of the parameters were based on values from the literature (Breuer et al., 2003). However, transpiration depends on the actual/potential evapotranspiration (or crop coefficient) and this value was calibrated by taking into account differences between land-use types from previous simulations (e.g. Bathurst et al. 2011, Birkinshaw et al. 2014).

The soil profile for each Shetran grid square comes from the 1km grid resolution HWSD database (www.fao.org/nr/water/news/soil-db.html (FAO/IIASA/ISRIC/ISSCAS/JRC, 2012). The Chinese data in this database comes from the Institute of Soil Science, Chinese Academy of Sciences which provided the recent 1:1,000,000 scale Soil Map of China. For each grid square the dataset gives the texture type of the topsoil (0-30cm) and, where it exists, the subsoil (30-100cm). This data was aggregated up to the 10km Shetran grid squares with the soil profile chosen being the most dominant in that square. Overall, this gave 930 soil profiles. Generally the higher elevation region has shallower soils and a sandy loam texture as opposed to a loam or clay loam texture in the lower elevation regions. Using the Hypres v2.0 database (Wösten et al. 1999), eusoils.jrc.ec.europa.eu/esdb_archive/esdbv2/fr_advan.htm, the top soil and subsoil textures were used to assign the Shetran soil parameters (porosity, residual moisture content, van Genuchten parameters and saturated hydraulic conductivity). There is little information available on the subsurface geology. Ge at al. (2008) provides some information for the soil and aquifer properties for the Tibetan plateau and there is some information available for the Sichuan basin (Li et al. 2007; Zhou and Li 1992). Li et al. (2007) note that the surface sediments in the Sichuan basin can produce an unconfined aquifer. Due to these uncertainties, where there is a subsoil, an aquifer is assumed within the model. The depth and hydraulic conductivity of the aquifer was calibrated, with the calibration carried out to produce a baseflow that corresponds with the measured discharges. A hydraulic conductivity value of 15m/day for a 4m deep aquifer produced the best fit.

Snow accumulation depends on both precipitation and air temperature with snowmelt calculated using a degree day method (as there was insufficient data to use the more complex energy budget methods) with the melt dependent on the sum of the positive air temperatures. Hock (2003) reviewed values for a variety of sites around the world and a typical value for snow of 3.5 mm d-1 $^{0}$C-1 was used here. Glaciers were not considered in this work as they make up less than 0.1% of the catchment (Immerzeel et al. 2010).

The remaining parameters that were calibrated were the Strickler overland flow coefficient (1.0) and the Strickler flow coefficient for the river channels (50.0). These affect the speed of surface water flow and so the shape of the hydrograph. A complete list of the calibrated parameters can be seen in Table 2.



A large number of dams exist within the Yangtze River basin upstream of the TGD (Yang et al. 2006). Due to the number of dams and the lack of knowledge of their operating procedures the dams are not simulated. However, with the large wet season precipitation totals, the dams seem to have little effect on the discharge at Yichang (see Sect. 3.1).

A standard split sample calibration/validation was carried out for the Shetran simulation. The calibration was for 1996-2000 and the validation period for 2001-2005. The comparison between measured and simulated discharge is made using the Nash Sutcliffe Efficiency (NSE).

## 2.3 CMIP5

We use outputs from Atmosphere-Ocean General Circulation Models (GCMs) from the fifth phase of the Climate Model Intercomparison Project (CMIP5) under RCP8.5. RCP8.5 has a rising pathway of radiative forcing of more than 8.5W/m2 in 2100 (more than 1370ppm CO2-equiv) (Moss et al. 2010). Sanderson et al. (2011) showed that RCP8.5 is similar to SRES A1FI (Nakićenović et al., 2000) and, although it is the highest emission scenario available in CMIP5, it still assumes emissions well below what the current energy mix would produce in the future.

Since no "general all-purpose metric" to identify best models exists (Knutti et al., 2010), we used all 78 CMIP5 runs for long term simulations under RCP8.5 available at http://climexp.knmi.nl at the time of download that contained both precipitation and air temperature. Table 3 details these experiments from 35 different GCMs, with several runs available for some of them. The downloaded CMIP5 outputs had been previously re-gridded and data was available for 21 grids ($2.5^0$ by $2.5^0$) within the Yangtze basin (shown in Fig. 1).

Due to their coarse resolution, and inability to resolve significant subgrid scale features, downscaling of GCM outputs is needed to assess local/regional impacts of climate change (Fowler et al., 2007). We use the simplest method: the change-factor (CF), perturbation or delta-change approach where the mean change between control and future GCM outputs is applied to daily observations (by adding or multiplying, depending on the variable in question). We analysed changes in precipitation and air temperature between 1981-2010 and 2041-2070 from 21 GCM grid cells over the Yangtze for each of the 78 CMIP5 runs, extracted monthly change factors (ratio for precipitation, absolute for temperature) and modified the observed time series data using the monthly CF from the nearest CMIP5 grid cell. There were 10 years of original data so the procedure gives 10 years of future precipitation and temperature data.

A CF method was also used to obtain future potential evapotranspiration (PET). PET for both historical and future periods was calculated from climate model temperature outputs using the Thornthwaite equation. With this, PET change factors for each CMIP5 model run and each grid were calculated (similar to the precipitation procedure). These CFs were then applied to the observed PET.

PET is a theoretical concept with inherent direct monitoring difficulties; several equations have been developed to calculate potential evapotranspiration from measurable variables. The reasons for using the Thornthwaite equation are considered in Sect. 4.2.



## 3 Results

### 3.1 Historical Discharge

Figure 4a shows the excellent match between the Shetran simulated and measured daily discharge at Yichang for monthly values from 1996-2005. The annual cycle of low discharges during the dry season (December to March) and then increasing

discharges up to July and then a gradual decrease back to December is well captured by the model, with only small differences between the measured and simulated values. The other plots in Fig. 4 compare measured and simulated mean daily discharges for two years of data. The daily discharge has a NSE of 0.95 for the calibration period (1996-2000) and 0.92 for the validation period (2001-2005). These NSE values are considerable higher than the value of 0.75 suggested by Moriasi et al. (2007) to class the simulation as 'very good'.

Figure 4e shows an obvious reduction in discharge at Yichang from 26/5/2003 – 12/6/2003. This reduction was due to the first impoundment of water in the dam, with water level at the TGD increasing from 65m to 135m a.s.l. (Wang et al. 2013). After this the water level remains fairly constant until the next impoundment in September 2006 (Wang et al. 2013). Therefore, as expected, the analysis of the discharge data at Yichang shows no obvious reduction for the rest of 2003-2005.

### 3.2 Comparison of CMIP5 Model Runs with Measured Data

Before considering future climate projections from the CMIP5 model runs it is first important to consider how well they predict the current climate. The ability of GCMs to capture the overall dynamics of the Asian summer monsoon is beyond the scope of this paper but work has previously been carried out by other researchers (Sperber et al, 2013; Song and Zhou, 2014; McSweeney et al. 2015; Dong et al. 2016). In Tables 3 and 4, we compare precipitation and temperature indices over the Yangtze basin for one run for each CMIP5 GCM (the other runs are shown in brackets) with those from observations.

The colouring indicates the quality of the model against observations using the same system as McSweeney et al. (2015) - 'Satisfactory' (green), 'Biases' (yellow), 'Significant biases' (orange) and 'Implausible' (red). In the second column of Table 3 we also reproduce results from McSweeney et al. (2015) to indicate the performance of the GCMs at reproducing large scale circulation flow at 850 hPa for the Asian summer monsoon. It can be seen that many of the models are poor in their simulation of the monsoon.

In Fig. 5 we consider model simulated precipitation in more detail. Figure 5a shows the large spread in annual precipitation amongst the models and that all CMIP5 model runs overestimate annual observed precipitation. The IPSL-CM5A-MR model is closest to the observed with a 30-year mean annual precipitation of 960.6mm and the worst the BNU-ESM model with 1919.5mm. All models also underestimate the fraction of precipitation occurring in summer (Fig. 5c). Together, the CMIP5 models give a multi-model ensemble average similar to the measured average during the wet season (although with a

large range) but considerably higher than observed precipitation during the dry season (Chen and Frauenfeld, 2014; Fig 5). However, the observed spatial distribution of precipitation across the Yangtze basin, from low precipitation in the Tibetan



Plateau to higher precipitation near the TGD, is captured better by most models. MIROC-ESM and MIROC-ESM-CHEM are poor, estimating nearly as much precipitation in the drier regions as in the wetter regions of the basin.

The results in Table 4 indicate that all CMIP5 GCMs underestimate observed mean annual temperature in the Yangtze basin (10.2°C). The MIROC5 model produces the best estimate, with a mean temperature of 7.3°C, and the worst is the CNRM-CM5 model with a mean value of 2.6°C. However, all models satisfactorily reproduce the observed spatial distribution and seasonality of temperature.

### 3.3 Future Changes

A majority of CMIP5 model runs (59 of the 78 models) predict increases in annual precipitation, with a smaller number (19) predicting decreases. Applying the non-parametric Kolmogorov-Smirnov (KS) test at the 0.05 significance level, indicates 44 models with a statistically significant increase in precipitation and 34 with no significant change. All the model runs predict statistically significant increases in temperature and potential evapotranspiration.

Considering the months separately, Figure 6 shows box plots of the spatially averaged changes in precipitation, temperature and potential evapotranspiration. Most models project increases in precipitation for all months, which can reach up to 40%, but some models project decreases in precipitation in some months. All models project increases in temperature in every month but this varies between just over 1°C to more than 4°C. Using the Thornthwaite equation changes in potential evapotranspiration are relatively small in winter because of the very low temperatures (mean Dec-Feb temperature is -5°C). However, in summer the projected increases in potential evapotranspiration are larger, with some models projecting increases up to 25mm.

Figure 7 shows box-plots of annual changes in precipitation, temperature and potential evapotranspiration for the different CMIP5 grid cells (i.e. the spatial variation). Most models project an annual increase in precipitation for all CMIP5 grid cells. Considering the median values and the percentage change in precipitation (Fig. 7b), the high and dry areas of the Tibetan plateau (grid cells 1, 2 and 3) show the biggest increases (10.8%, 8.6% and 9.4%) and the areas furthest south (grid cells 7,11 and 15) show the smallest increases (2.8%, 0.91% and 1.4%). However, the projections show a wide range, with individual models indicating both increases and decreases in annual precipitation in all areas of the basin. Considering the change in precipitation (Fig 7a), the range of possible changes from CMIP5 is largest in the south-western part of the basin (grids 4, 7,8, 11 and 12) and this uncertainty will have an important effect on the future volume of discharges in the Yangtze river at the TGD. The high altitude areas (grids 1, 2, 3, 5, 6, 9 and 10) show the largest temperature increases (Fig. 7c) but, due to their current low temperatures, small increases in potential evapotranspiration (Fig. 7d). Accordingly, the warmer eastern areas show a higher increase in potential evapotranspiration as a $2^0$C rise in air temperature has a larger effect at higher temperatures.

Figure 8 shows box plots of future projections (2041-2070) for the basin's annual average precipitation, potential evapotranspiration, simulated discharge and simulated actual evapotranspiration from the 78 CMIP5 runs. The blue squares



show the values for the present climate. Most models runs project increases in precipitation and all models show an increase of potential and consequently actual evapotranspiration (since water availability during the warm season is not an issue). These two factors combined mean that the spread of future discharge projections for the annual totals encompass the present conditions, with 11 model runs showing an increase and 67 a decrease in annual discharge.

Future discharge projections for individual months for all 78 future climate runs are shown as box plots in Fig. 9, with the current climate shown as a blue square. Current discharges are encompassed in the inter-model spread for the future for all months. However, most models show a decrease in discharge in every month compared to the current climate, with the largest decreases in the wet season. A reduction in discharge early in the wet season would affect agricultural production within the Yangtze basin and a reduction in discharge late in the wet season (September and October) would affect hydro-

power production in the dry season at TGD since these are particularly important months for filling its reservoir. The largest reduction in projected discharge is in June (with 72 models showing a decrease in discharge and 6 an increase), partly due to changes in snow accumulation and melt. The modelling suggests that under the present climate 4.2mm of June discharge is from snowmelt; this reduces to 2.2mm for the median of the CMIP5 simulations. June discharge is also affected by higher evapotranspiration in the CMIP5 simulations compared to the present day climate as the earlier snowmelt allows the

evapotranspiration to start earlier (the snow covering the vegetation prevents any evapotranspiration).

We plot the change in both precipitation and simulated discharge between current and future projections for each of the 78 CMIP5 runs in Fig. 10a. The multi-model mean increase in precipitation is 4.1% which corresponds to an 11.1% decrease in discharge (shown by the red square). The 78 CMIP5 runs show a large range of potential future outcomes: from a 3.6% drop to a 14.8% increase in precipitation and a 29.8% drop to a 16.0% increase in discharge. The slope of the fitted line through

all 78 CMIP5 runs in Fig. 10a shows that a 10% change in precipitation produces, on average, an 18.7% change in annual discharge. The problem with this analysis is that some of the GCMs have multiple runs (e.g. CSIRO-Mk3-6-0 has 10) and these are not independent. Therefore, in Fig. 10b, the simulated discharge between current and future projections for each of the 35 GCMs are plotted (the individual models are also labelled). This shows very similar results, with a multi- model mean increase in precipitation of 3.9% which corresponds to an 11.9% decrease in discharge. The range of precipitation (-3.5 to +

13.6%) and discharge (-29.8 to +7.0%) is slightly reduced. The colours in Fig. 10b correspond to those used by McSweeney et al. (2015) to assess the performance of models at reproducing the climate of the Asian summer monsoon (see Table 3). The 'satisfactory' green points cover almost the entire range and so it is very hard to discount any future projections of change in precipitation or discharge.

## 3.4 Comparison of the CanESM2 and CSIRO-Mk3-6-0 Models

To understand why there is such a large range in the future projections of discharge in the Yangtze basin, two models were selected and analysed in more detail: CanESM2 and CSIRO-Mk3-6-0. Both models are able to represent the large-scale circulation of the Asian summer monsoon satisfactorily (McSweeney et al. 2015) and are two of the best models at simulating precipitation and temperature indices in the Yangtze Basin (see Tables 3 and 4). However, although both models





project similar increases in temperature, 3.05°C and 2.85°C, respectively, CanESM2 projects a 12.7% increase in precipitation and a 2.6% increase in discharge, whereas CSIRO-Mk3-6-0 projects a 3.1% decrease in precipitation and a 26.3% decrease in discharge. Figure 11 a-c shows the annual, monthly and spatial precipitation variability over the Yangtze basin in both models. Both show a slightly earlier onset of the summer monsoon (Fig. 11b) in a future climate but the key

difference between the two can be seen in Fig. 11c which considers the distribution of precipitation across the CMIP5 grid cells. CanESM2 shows a very large increase in annual precipitation in grid 4 (604mm) and also large increases of more than 250mm in grid cells 5, 7, 8 and 9, whereas, CSIRO-Mk3-6-0 projects no significant change or a slight reduction in precipitation in these grids. These spatial differences in precipitation produce the large difference in the projected discharge, seen for each month in Fig. 11d. Grid cells 4, 5, 7, 8 and 9 (Fig. 1) are in the south-west part of the Yangtze basin and, as

shown in Fig. 7, show the greatest range in future projections across all the models. So most of the variation in discharge across the different models is due to the change in precipitation seen across these grids in the south-west of the basin. This is considered further in Sect. 4.4.

## 4. Discussion

### 4.1 Change Factor Approach

In this paper the change-factor (CF), perturbation or delta-change approach was used to produce the future climate scenarios. The simplicity of this method makes it possible to downscale several GCMs/scenarios quickly but on the other hand it assumes that the GCM bias is constant and that variability, spatial patterns of climate and percentage of wet/dry days will remain constant (Fowler et al., 2007). However this method does preserve the observed spatial correlations between stations or grid points, which some complex methods are not able to do and it also captures the full climate signal of the GCM, while

more complex downscaling methods capture only climate forcing shown by the chosen predictor(s) and grid box(es) (Diaz-Nieto and Wilby 2005). The CF method is not suitable for the study of extreme events (since it does not take into consideration any changes to the variance and skewness of the precipitation). For small river basins this might have significant consequences in the projected discharge; however, in a large river basin such as the Yangtze there is considerable attenuation of the hydrograph. Therefore, the consequences of changes in the precipitation variance and skewness on the

basin's monthly mean discharges will be much smaller. Also, this is a widely used method that has been considered appropriate for studies where changes in average values are relevant such as impacts on water resources (Sunyer et al. 2010).

### 4.2 Potential Evapotranspiration (PET)

There is still considerable debate about the best method for calculating PET under a changing climate using climate model outputs, see for e.g. Ekström et al. (2007), Kingston et al. (2009), Weiland et al. (2012), Prudhomme and Williamson (2013)

and McMahon et al. (2015). In this study the Thornthwaite equation was chosen to calculate change factors because, although simplistic, it only requires temperature time-series which is a fairly reliable GCM output. The Penman-Monteith



method was not used as it is based on variables that are not well simulated by GCMs, like cloud cover and vapour pressure (Kingston et al. 2009) and the results can be physically unrealistic (Ekström et al. 2007). However, this does mean that changes in PET as a result of changes in wind speed, cloud cover and vapour pressure deficit are not accounted for and Chen et al. (2006) and Yan et al. (2011) have shown the importance of these meteorological variables for PET from the Tibetan

Plateau. McMahon et al. (2015) suggest that the effect of using just temperature to calculate PET is likely to be most important in an energy limited regions such as the Tibetan plateau but is less important in other regions such as the rest of the Yangtze basin.

Nevertheless, as a sensitivity test, the model experiments were run for a second time with no change to PET (i.e. the time series of ten years of historic PET data was used together with the projected changes in precipitation and temperature). This

produced an inter-model basin mean increase in discharge of 7.5% with a range between -7.6% and +28.7%. As expected, the discharges are significantly higher than those using future PET where the inter-model basin mean reduction in discharge was 11.1% with a range between -29.8% and +16.0%. This shows the relevance of the changes in PET for future discharges and the need for future research on how to calculate realistic PET from climate model outputs.

### 4.3 Limited measured data

Only ten years of meteorological data was used to calibrate and validate the hydrological model of the Yangtze, with the comparison of the measured and simulated discharge data showing that the Moriasi et al. (2007) 'very good' criteria value was easily exceeded in every year. Ideally a longer time series of measured meteorological and discharge data would be available so any annual extremes or inter-decadal variation in precipitation can be captured by the model. To test the effects of using only 10 years of measured precipitation data, the areal averaged annual totals (from 64 stations) were compared

against the Global Precipitation Climate Centre (GPCC) dataset. Gridded GPCC data is available for monthly precipitation at 0.5° x 0.5° from 1901 -2010 (Schneider et al. 2013), and is based on in-situ observations across global land areas.  Figure 5a shows that precipitation totals from GPCC are consistently slightly lower than the data observations used in this study. However, both show similar inter-annual variability .In the 30-year GPCC record there are no extremes of precipitation which are large outliers to the 10-years of precipitation observations used in this study.

There are still uncertainties in using Shetran to predict discharges for precipitation outside the limits of the model calibration and validation period. However, as Shetran is a physically-based model, theoretically this means that the predicted discharges will be representative of future climates. In fact, Sect. 3.1 and 3.2 show that under the current climate the ability of Shetran to simulate the discharge is considerably better than to the ability of the CMIP5 model runs to produce a good correspondence with precipitation indices. So it could be argued that under a future climate the uncertainties in discharge

from using Shetran are smaller than the uncertainties in the projected future climate.





## 4.4 Climate Change

Using 78 climate projections under RCP8.5 from the most recent generation of climate models (CMIP5), the analysis shows that between 1981-2010 and 2041-2070 projections of change to basin annual precipitation vary from -3.6% to +14.8%, with a multi-model mean of 4.1%. This small increase in precipitation agrees with other analyses of projected changes to precipitation from both the previous generation of climate models (CMIP3) and the most recent ones (CMIP5) (Piao et al., 2010; Tao et al. ,2012; Jiang and Tian, 2013; Tian et al., 2015).

However, in this study we focus on the changes to discharge, using projections from 78 CMIP5 model runs together with a hydrological model. Overall, a multi-model basin mean reduction in discharge of 11.1% was projected for 2041-2070, with a range between -29.8% and +16.0%. The results suggest no agreement in the sign of change and a potentially large range of values.

The key to predicting future changes to discharge in the Yangtze basin is correctly predicting how the strength and location of the summer monsoon will change under a future climate. Lee and Wang (2014) evaluated 20 CMIP5 models while considering future changes in the monsoon and selected the four best ones, which included the CanESM2 model. The four 'best' models projected that the land monsoon domain over Asia will expand westward with a 10.6 %.increase in monsoon extent under the RCP4.5 scenario. However, we have shown that there is major uncertainty in this supposed expansion into the Yangtze basin, as in some GCMs (e.g. CanEMS2) there is an expansion of the monsoon domain north and west and this increased precipitation produces an increase in the discharge, whereas, in most other GCMs (e.g. CSIRO-Mk3-6-0) there is not an expansion in the domain and so it results in a decrease in the discharge (due to greater evapotranspiration). Until the strength and location of the monsoon under a future climate can be reliably predicted there will remain large uncertainty in changes to projected discharge for the Yangtze basin.

## 5. Conclusions

Water resources, flooding and hydro-power generation on the Yangtze River are all important due to the size of the population and the industry and agriculture it supports. Variability in the Yangtze discharge under a future climate is therefore of great concern. This study has, for the first time, taken 78 state-of-the-art climate model projections from CMIP5 (from 35 different GCMs) and used these together with a detailed hydrological model of the Yangtze basin to estimate potential changes to future discharge.

We considered 78 CMIP5 projections for the Yangtze RCP8.5 and examined the change in precipitation between 1981-2010 and 2041-2070. The results showed a big spread, without agreement even in the sign of the change for both monthly and annual precipitation (from -3.6% to +14.8%).  However, most GCMs projected an increase in precipitation for most months with a multi-model basin mean change of +4.1%. GCM projections for change in temperature for the same time period showed significant increases, which varied from just over 1°C to more than 4°C. The changes in potential evapotranspiration, calculated using the Thornthwaite equation, also showed significant increases.





The Shetran hydrological model gave an excellent match between measured and simulated discharge for the Yangtze River basin to Yichang (1,007,200 km2) with a Nash-Sutcliffe efficiency of 0.95 for the calibration period in 1996-2000, and of 0.92 for the validation period from 2001-2005. Using monthly change factors within the basin to modify the historic meteorological data, future climate scenarios were obtained for each of the 78 CMIP5 projections and applied to Shetran.

These produced a multi-model basin mean change in discharge of -11.1%, with a range between -29.8% and +16.0%.

Overall, this work has highlighted the uncertainty in GCM projected changes of precipitation and temperature and their effect on the discharge in the Yangtze basin. In particular, it has highlighted the importance of predicting the strength and location of the summer monsoon. To fully understand the effect that climate change will have on the Yangtze basin, there needs to be an improvement in climate model projections and, in particular, of precipitation over the basin. Piao et al. (2010)

came to a similar conclusion looking at the effect of climate change on agriculture in China.

An improvement in climate model performance would also allow potential evapotranspiration to be calculated using the Penman-Monteith equation which would deliver more reliable projections of change in potential evapotranspiration. Further work is also needed looking at how changes in extreme precipitation can cause floods and we intend to carry out future Shetran simulations using different downscaling techniques.

**Acknowledgements**

This work was a result of the following funding projects: (1) 973 Program (Grant no. 2013CB036401); (2) UK EPSRC Global Secure project (EP/K004689/1). H.J.F. was supported by the Wolfson Foundation and the Royal Society as a Royal Society Wolfson Research Merit Award holder (WM140025). We acknowledge the World Climate Research Programme's Working Group on Coupled Modelling, which is responsible for CMIP, and thank the climate modelling groups for

producing and making available their model output. We would also like to thank Dr Geert Jan van Oldenborgh from the KNMI Climate Explorer website for making outputs from CMIP5 available in an easy to use and downloadable format.



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



**Table 1.** Vegetation parameters used in the Shetran simulations of Yangtze. [1]The leaf area index values vary seasonally. [2]The Actual /Potential evapotranspiration is the value at field capacity it reduces as the soil dries (this parameter was calibrated).

| Vegetation type | Leaf area index[1] | Canopy storage capacity (mm) | Actual /Potential evapotranspiration[2] |
|---|---|---|---|
| Cropland | 0.01-1.0 | 1.5 | 0.8 |
| Shrub/herbaceous | 0.01-0.8 | 1.0 | 0.6 |
| Evergreen forest | 0.5-1.0 | 3.0 | 1.0 |
| Deciduous forest | 0.1-1.0 | 3.0 | 1.0 |
| Bare rock/soil | 0.0 | 0.2 | 0.5 |
| Rice paddies | 0.01-1.0 | 1.5 | 0.8 |
| Lake/wetlands | 0.0 | 0.0 | 1.0 |



**Table 2.** List of parameters calibrated in the Shetran model during the calibration period (1996-2000) - see the text for details. The Strickler coefficient is the inverse of the Manning coefficient

| Parameter |
| --- |
| Actual/potential evapotranspiration for each vegetation type |
| Aquifer depth |
| Aquifer conductivity |
| Strickler overland flow coefficient |
| Strickler flow coefficient for river channels |





**Table 3.** CMIP5 model run performance summary for precipitation. The results from 78 runs simulated using 35 GCMs are shown; if a model has several different runs the numbers of runs and the ranges are shown in brackets. The summer monsoon results are from McSweeney et al. (2015). The performance is based on the ability of the GCMs at reproducing large scale circulation flow at 850 hPa for the Asian summer monsoon and is identified as 'Satisfactory' (green), 'Biases' (yellow), 'Significant biases' (orange) and 'Implausible' (red). The other results are calculated in this study for the Yangtze basin. The colours show the comparison with the measured data with green being the best through yellow, orange and red the worst. The measured data is for 10 years (1996-2005) whereas the CMIP5 data is for 30 years (1980-2009), see Fig. 5.

| Model | Summer Monsoon | Mean (°C) | Spatial distribution T90-T10 (°C) | Intra-annual distribution. Dry months mean - DJF (°C) | Intra-annual distribution. Wet months mean - JJA (°C) | Inter-annual variation. Minimum mean (°C) | Inter-annual variation. Maximum mean (°C) | Future Change (°C) |
|---|---|---|---|---|---|---|---|---|
| **Measured** | | **904.3** | **100.0** | **3.6** | **54.8** | **815.0** | **1008.0** | |
| ACCESS1-0 | | 1307.8 | 60.4 | 8.2 | 43.7 | 1205.2 | 1406.6 | 9.8 |
| ACCESS1-3 | | 1607.2 | 63.7 | 8.6 | 44.0 | 1422.1 | 1783.8 | 8.2 |
| bcc-csm1-1 | | 1537.2 | 46.1 | 10.9 | 41.8 | 1317.7 | 1828.8 | 1.4 |
| BNU-ESM | | 1919.5 | 54.1 | 11.8 | 39.0 | 1735.4 | 2157.2 | -0.4 |
| CanESM2 (5) | | 1275.7 (1249-1297) | 108.6 | 6.8 | 46.4 | 1104.3 | 1494.6 | 12.7 (9.8 - 14.9) |
| CCSM4 (6) | | 1273.1 (1271-1316) | 71.8 | 6.0 | 46.7 | 1160.1 | 1442.1 | 6.5 (-0.5 – 6.5) |
| CESM1-BGC | | 1280.6 | 72.6 | 6.4 | 45.8 | 1169.3 | 1405.3 | 6.4 |
| CESM1-CAM5 (3) | | 1430.9 (1426-1440) | 65.5 | 6.6 | 46.1 | 1231.9 | 1573.8 | 10.9 (10.9 - 14.3) |
| CMCC-CM | | 1121.3 | 66.8 | 8.5 | 44.4 | 974.3 | 1305.9 | 1.0 |
| CMCC-CMS | | 1298.5 | 84.2 | 9.1 | 44.1 | 1125.8 | 1495.6 | 3.4 |
| CNRM-CM5 (5) | | 1343.7 (1303-1348) | 102.2 | 11.0 | 44.0 | 1159.3 | 1472.6 | 2.8 (2.3 - 6.7) |
| CSIRO-Mk3-6-0 (10) | | 1233.2 (1214-1234) | 121.1 | 5.8 | 48.9 | 1066.5 | 1397.7 | -3.1 (-3.5 - 2.8) |
| EC-EARTH (4) | | 1105.5 (1090-1106) | 89.2 | 8.7 | 42.8 | 989.6 | 1210.6 | 0.1 (0.1 - 4.5) |
| FGOALS_g2 | | 1146.1 | 57.7 | 6.5 | 40.8 | 995.1 | 1275.9 | -0.1 |
| FIO-ESM (3) | | 1880.7 (1867-1883) | 75.8 | 10.2 | 41.1 | 1653.5 | 2083.3 | -3.5 (-3.5 - 3.0) |
| GFDL-CM3 | | 1442.8 | 61.4 | 9.4 | 44.0 | 1260.8 | 1689.0 | 8.0 |
| GFDL-ESM2G | | 1400.1 | 80.2 | 8.6 | 47.1 | 1186.5 | 1576.4 | 3.6 |
| GFDL-ESM2M | | 1495.9 | 75.2 | 8.5 | 45.5 | 1329.5 | 1652.6 | -0.5 |
| GISS-E2-H (p1-p3) | | 1389.9 (1355-1490) | 146.1 | 9.9 | 37.8 | 1231.7 | 1495.5 | 0.6 (0.6 - 2.3) |
| GISS-E2-R (r1-r3) | | 1324.4 (1249-1436) | 130.8 | 12.1 | 38.2 | 1203.8 | 1447.4 | -1.3 (-1.6 - 4.6) |
| HadGEM2-AO | | 1279.1 | 59.4 | 7.6 | 44.5 | 1174.6 | 1416.4 | 13.6 |
| HadGEM2-CC | | 1265.3 | 59.0 | 7.9 | 44.0 | 1181.1 | 1404.2 | 7.5 |





| | | | | | | | |
|---|---|---|---|---|---|---|---|
| HadGEM2-ES (4) | 1253.7 (1226-1254) | 55.0 | 8.3 | 44.4 | 1149.4 | 1328.9 | 6.2 (5.6 – 8.1) |
| inmcm4 | 1495.9 | 85.6 | 8.8 | 45.9 | 1310.7 | 1648.1 | -1.2 |
| IPSL-CM5A-LR (4) | 1078.9 (1079-1104) | 128.9 | 9.5 | 41.7 | 909.7 | 1223.8 | 2.8 (2.2 – 4.1) |
| IPSL-CM5A-MR | 960.6 | 90.2 | 7.3 | 46.4 | 748.3 | 1088.2 | -2.7 |
| IPSL-CM5B-LR | 1197.6 | 136.8 | 9.9 | 39.8 | 947.0 | 1351.1 | 1.9 |
| MIROC5 (3) | 1403.2 (1403-1418) | 77.1 | 5.0 | 45.3 | 1290.1 | 1533.5 | 12.4 (12.0 – 12.4) |
| MIROC-ESM | 1129.2 | 37.3 | 4.3 | 48.9 | 934.5 | 1343.1 | 9.0 |
| MIROC-ESM-CHEM | 1166.6 | 37.9 | 4.7 | 48.3 | 1040.8 | 1353.9 | 6.9 |
| MPI-ESM-LR (3) | 1374.0 (1366-1374) | 87.9 | 7.7 | 44.5 | 1256.3 | 1515.8 | -0.8 (-0.8 – 0.6) |
| MPI-ESM-MR | 1334.6 | 92.4 | 7.9 | 45.8 | 1172.9 | 1440.1 | 3.4 |
| MRI-CGCM3 | 1027.1 | 85.2 | 8.7 | 43.3 | 922.6 | 1163.6 | 0.3 |
| NorESM1-M | 1634.8 | 74.7 | 5.6 | 47.1 | 1454.6 | 1812.4 | 4.7 |
| NorESM1-ME | 1590.5 | 75.1 | 5.8 | 46.3 | 1433.6 | 1771.9 | 6.0 |



**Table 4.** CMIP5 model run performance summary for temperature. As in Table 3 the results from 78 runs simulated using 35 GCMs are shown; if a model has several different runs the numbers of runs and the ranges are shown in brackets. The colours show the comparison with the measured data with green being the best through yellow, orange and red the worst. For all CMIP5 models the spatial distribution and intra-annual distribution are considered to be satisfactory. The measured data is for 10 years (1996-2005) whereas the CMIP5 data is for 30 years (1980-2009), see Fig. 5.

| Model | Mean (°C) | Spatial distribution T90-T10 (°C) | Intra-annual distribution. Dry months mean - DJF (°C) | Intra-annual distribution. Wet months mean - JJA (°C) | Inter-annual variation. Minimum mean (°C) | Inter-annual variation. Maximum mean (°C) | Future Change (°C) |
|---|---|---|---|---|---|---|---|
| **Measured** | **10.24** | **21.5** | **1.34** | **18.21** | **9.64** | **10.69** | |
| ACCESS1-0 | 5.39 | 17.87 | -5.27 | 14.94 | 4.67 | 6.39 | 3.13 |
| ACCESS1-3 | 6.01 | 16.37 | -3.69 | 15.52 | 5.27 | 6.67 | 2.71 |
| bcc-csm1-1 | 5.15 | 17.69 | -5.56 | 14.76 | 4.29 | 6.47 | 2.39 |
| BNU-ESM | 6.28 | 17.67 | -4.39 | 16.41 | 5.79 | 7.03 | 2.78 |
| CanESM2 (5) | 6.84 (6.83 - 6.91) | 18.36 | -4.45 | 17.01 | 5.58 | 8.40 | 3.05 (3.05-3.45) |
| CCSM4 (6) | 5.20 (5.12 - 5.27) | 20.23 | -6.06 | 15.39 | 4.23 | 6.23 | 2.49 (2.32.-2.54) |
| CESM1-BGC | 5.10 | 20.32 | -5.98 | 15.39 | 4.55 | 6.28 | 2.34 |
| CESM1-CAM5 (3) | 4.72 (4.61 - 4.72) | 21.17 | -6.30 | 14.31 | 4.16 | 5.52 | 2.93 (1.56-2.96) |
| CMCC-CM | 5.51 | 18.93 | -4.17 | 13.60 | 4.73 | 6.32 | 2.71 |
| CMCC-CMS | 6.45 | 17.57 | -2.98 | 14.44 | 5.58 | 7.68 | 2.91 |
| CNRM-CM5 (5) | 2.61 (2.61 - 2.68) | 22.57 | -8.09 | 12.65 | 1.88 | 3.76 | 2.13 (1.82-2.13) |
| CSIRO-Mk3-6-0 (10) | 6.19 (6.17-6.36) | 20.12 | -4.79 | 16.29 | 5.57 | 7.32 | 2.85 (2.65-2.87) |
| EC-EARTH (4) | 4.64 (4.58 - 4.68) | 20.23 | -4.55 | 12.95 | 3.76 | 5.46 | 2.24 (2.12-2.24) |
| FGOALS_g2 | 4.20 | 22.32 | -6.77 | 14.27 | 3.79 | 5.24 | 2.24 |
| FIO-ESM (3) | 7.17 (7.17 - 7.25) | 17.25 | -2.64 | 16.23 | 6.44 | 8.91 | 2.57 (1.68-2.57) |
| GFDL-CM3 | 4.47 | 19.77 | -5.91 | 13.42 | 3.01 | 6.06 | 1.67 |
| GFDL-ESM2G | 5.57 | 18.71 | -3.17 | 13.89 | 5.00 | 6.55 | 3.69 |
| GFDL-ESM2M | 5.47 | 18.54 | -3.25 | 13.93 | 4.87 | 6.73 | 1.89 |
| GISS-E2-H (p1-p3) | 4.16 (4.00 - 5.42) | 25.71 | -4.58 | 12.28 | 3.27 | 5.12 | 2.08 (2.08-2.78) |



| | | | | | | |
|---|---|---|---|---|---|---|
| GISS-E2-R (r1-r3) | 5.25 (5.18 – 5.91) | 23.14 | -2.81 | 13.03 | 4.66 | 6.12 | 2.75 (2.27-2.75) |
| HadGEM2-AO | 5.64 | 18.23 | -5.42 | 15.52 | 4.60 | 6.51 | 2.68 |
| HadGEM2-CC | 4.61 | 16.99 | -6.64 | 14.69 | 4.09 | 5.46 | 2.64 |
| HadGEM2-ES (4) | 5.32 (5.29 – 5.38) | 16.67 | -5.42 | 15.15 | 4.10 | 6.24 | 3.51 (3.15-3.51) |
| inmcm4 | 2.98 | 22.06 | -9.59 | 13.94 | 2.14 | 4.18 | 3.25 |
| IPSL-CM5A-LR (4) | 4.20 (4.20 – 4.36) | 24.06 | -5.85 | 14.34 | 2.96 | 5.61 | 1.65 (1.65-3.54) |
| IPSL-CM5A-MR | 5.80 | 23.81 | -3.78 | 15.58 | 4.82 | 7.45 | 3.58 |
| IPSL-CM5B-LR | 3.43 | 22.72 | -7.41 | 13.42 | 2.36 | 4.52 | 3.44 |
| MIROC5 (3) | 7.29 (7.29 – 7.37) | 21.28 | -2.73 | 16.41 | 6.58 | 8.30 | 2.19 (2.19-2.94) |
| MIROC-ESM | 5.20 | 26.08 | -5.86 | 15.31 | 4.44 | 6.54 | 2.85 |
| MIROC-ESM-CHEM | 4.98 | 26.24 | -6.12 | 15.06 | 4.64 | 5.87 | 3.29 |
| MPI-ESM-LR (3) | 6.39 (6.30-6.39) | 18.52 | -2.87 | 14.45 | 5.96 | 7.47 | 3.68 (2.60-3.68) |
| MPI-ESM-MR | 6.50 | 19.07 | -3.01 | 14.68 | 5.70 | 7.58 | 2.73 |
| MRI-CGCM3 | 3.88 | 20.54 | -7.48 | 14.52 | 3.43 | 4.93 | 2.60 |
| NorESM1-M | 4.55 | 18.87 | -6.70 | 14.54 | 3.86 | 5.33 | 2.24 |
| NorESM1-ME | 4.49 | 18.88 | -6.64 | 14.50 | 3.80 | 5.27 | 2.55 |



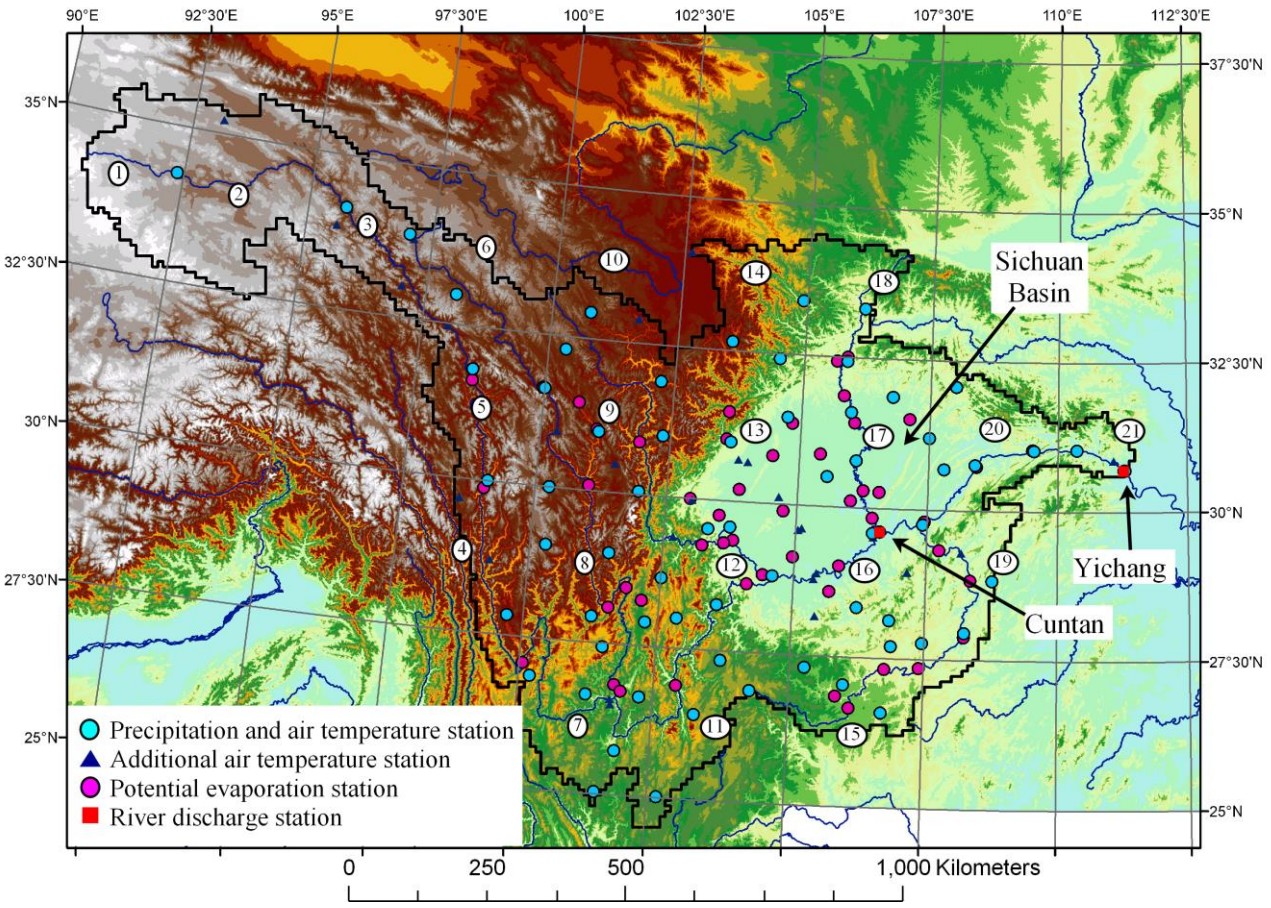

**Figure 1.** Yangtze River basin to Yichang (1,007,200 km2). Elevations range from over 5000m at the source of the Yangtze in the Tibetan plateau in the west to 65m at Yichang. CMIP5 outputs are available for each of the $2.5^0$ by $2.5^0$ grids numbered in the figure. In total 21 CMIP5 grids within or close to the Yangtze basin were used in this study. 64 precipitation and air temperature stations are shown and also shown are the locations of an additional 26 air temperature stations where there is not precipitation data (90 in total). The locations of the 52 potential evaporation stations are shown.




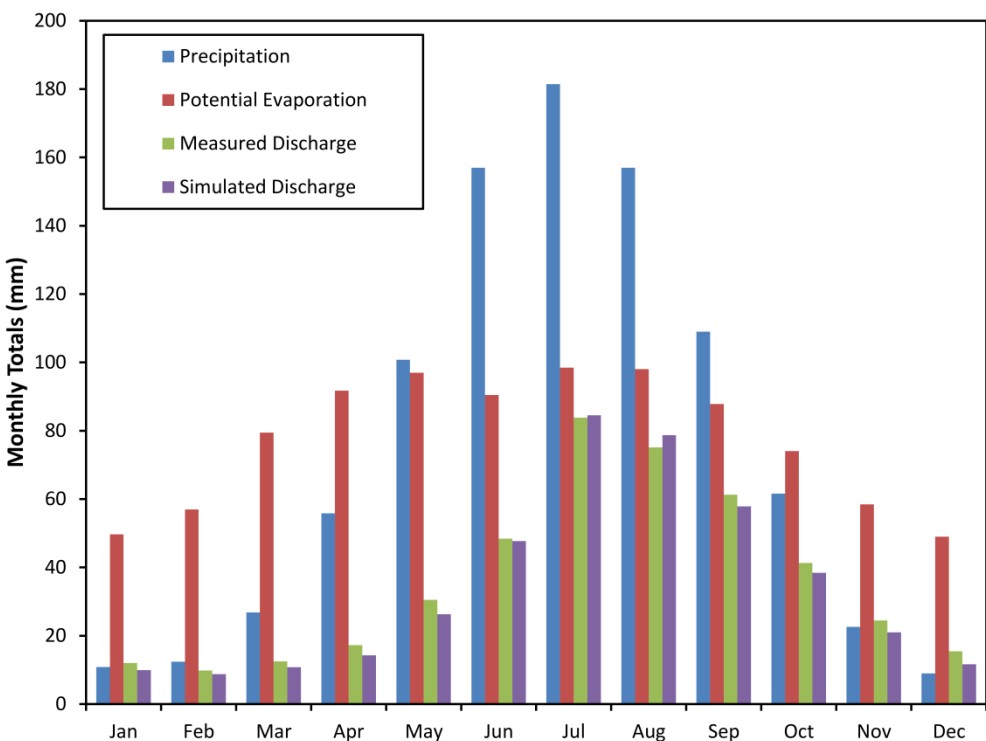

**Figure 2.** Monthly precipitation, potential evaporation, measured and simulated discharge totals in the Yangtze basin to Yichang, 1996-2005.





**Figure 3. (a)** Theissen polygon annual precipitation totals over the Yangtze basin. The high value (gauge 56385) at the western edge of the Sichuan Basin is discussed in the text. **(b)** Theissen polygon mean annual air temperature over the Yangtze basin.



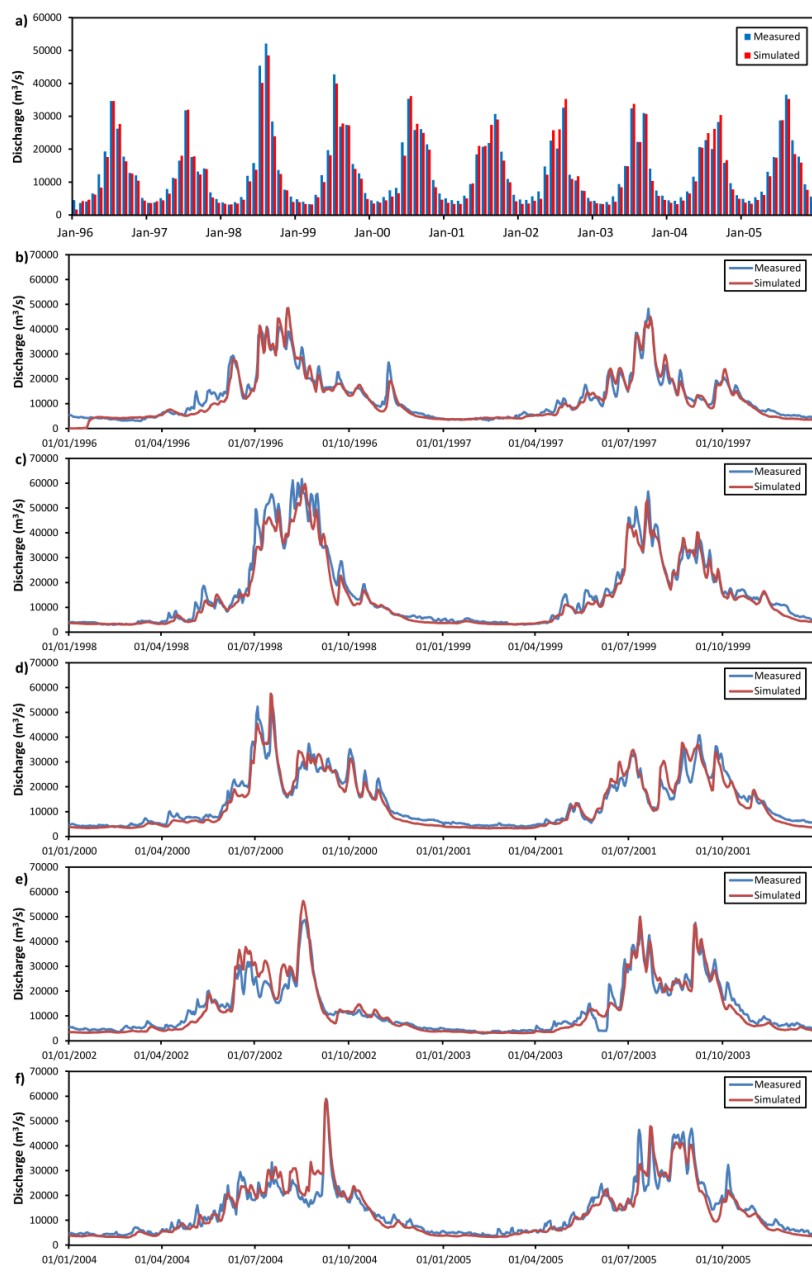

**Figure 4.** Measured and simulated daily discharges for the Yangtze at Yichang from 1996-2005. **(a)** monthly averages **(b)-(e)** daily averages for two year periods.



**Figure 5.** Comparison of the measured and CMIP5 climate models for the precipitation in the Yangtze basin. Only one run from each GCM is shown. **(a)** Annual precpitation (measured GPCC data is also shown), **(b)** Spatial distribution across the Yangtze basin showing the 10th - 90th percentiles  **(c)** intra-annual  distribution. The continuous lines are those that have the best correspondence with the measured data and the dashed the worst (see Table 3 for details).





**Figure 6.** Box plots showing monthly changes between 1981-2010 and 2041-2070 in precipitation, temperature and potential evaporation for the 78 CMIP5 runs averaged over the Yangtze basin.







**Figure 7.** Box plots showing spatial changes between 1981-2010 and 2041-2070 in precipitation, temperature and potential evaporation for the 78 CMIP5 runs. The numbers on the x-axis correspond to the 2.5⁰ by 2.5⁰ CMIP5 grids numbered in the Fig 1.



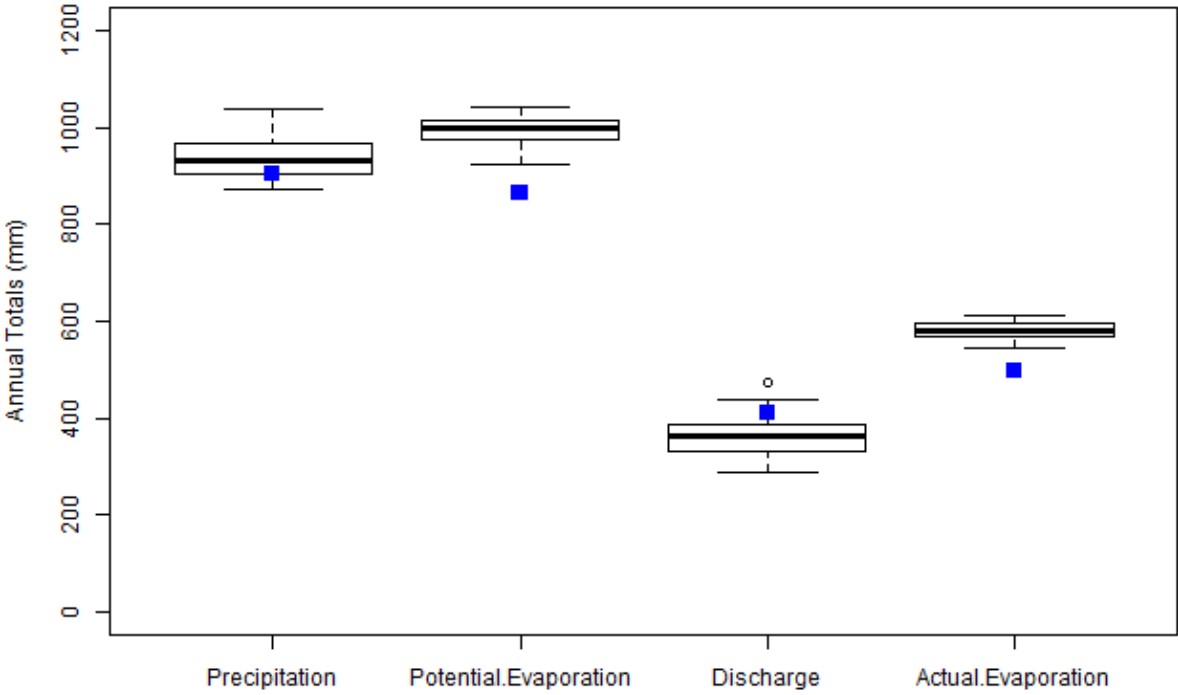

**Figure 8.** Box plot showing the range of precipitation, potential evaporation, simulated discharge and simulated actual evaporation over the 78 CMIP5 future climates. The simulated discharge for the current climate is shown by the blue square.




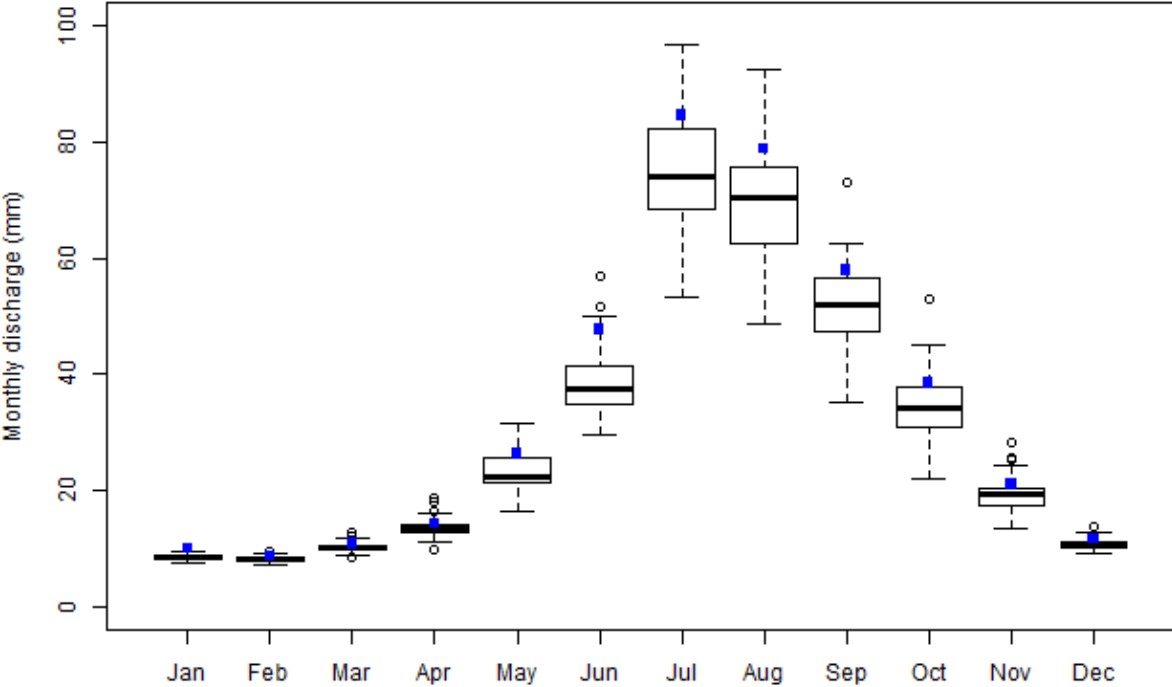

**Figure 9.** Box plots for individual months showing the range of simulated discharges over the 78 CMIP5 future climates. The simulated discharge for the current climate for each month is shown by the blue square.





**Figure 10.** Change in simulated discharge and precipitation between the current and future climates. **(a)** for each of the 78 CMIP5 future climate projections. **(b)** for each of the 35 GCM models (labelled). The colours correspond to those in Table 3 for the summer monsoon whereby green is 'satisfactory', yellow 'biases', orange 'significant biases', red 'implausible' and grey models do not have lateral boundary conditions available.




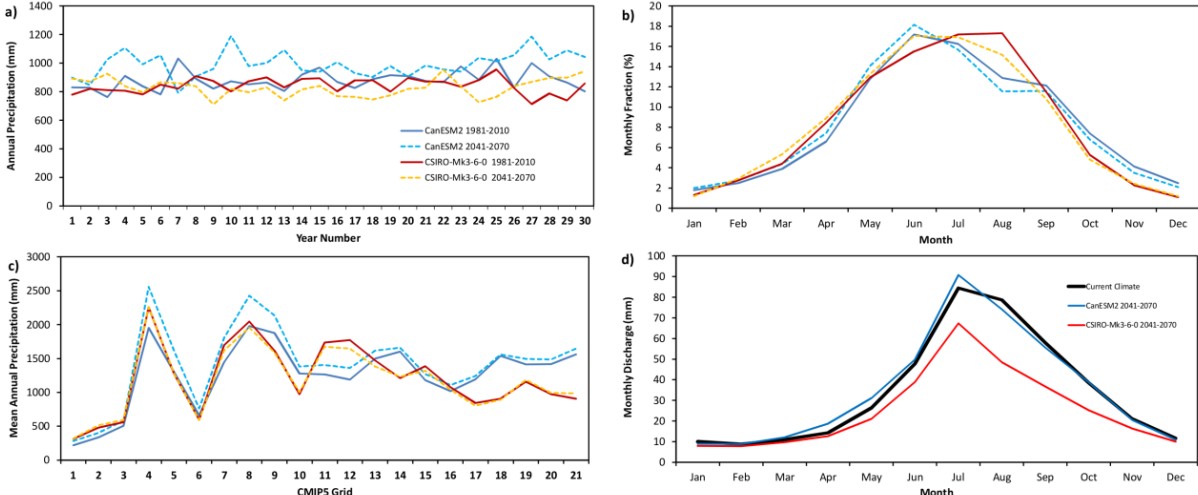

**Figure 11.** Comparison of the current climate and future climate projections for the CanESM2 model and the CSIRO Mk3-6-0 model. **(a)** Annual precipitation, **(b)** Monthly precipitation fraction **(c)** Mean annual precipitation for each $2.5^0$ by $2.5^0$ CMIP5 grid numbered in the Fig 1, **(d)** Monthly discharge from using the Shetran hydrological model

