# Peer review of "Climate Change Impacts on Yangtze River Discharge at the Three Gorges Dam"

_Hydrology and Earth System Sciences, 2016_

## Referee Comment (RC1) · Anonymous Referee #1 · 26 Jun 2016

This paper describes a study on climate impact on Yangtze River Discharge at the Three Gorges Dam. The topic is relevant to the journal's remit. The use of the high resolution Shetran physically-based distributed hydrological model represents a different model to other studies in the same region. The comparison between the CanESM2 and CSIRO-Mk3-6-0 Models is quite interesting. However, there are some issues that should be improved or clarified.

My main concern is on the short study period for the river discharge (only ten years). For climate studies, a ten year period cannot show clear patterns of climate change and environmental change (land use/land cover). Yangtze River is a major river in China and it is a surprise that the authors couldn't find more flow records beyond the ten year study period. Some other minor issues 1) More justifications/exploration on the Shetran model would be useful. As the authors have said 'By Other hydrological

models have previously been applied to the Yangtze basin (Hayashi et al. 2008, Woo et al. 2009, Xu et al. 2008), but in terms of grid resolution, this is the most detailed hydrological model that has been produced for a major part of this basin.' It would be useful if the authors could try different model resolutions to show the resolution effect on the modelling result at this basin so that a high resolution model is justified. A comparison with other models from the aforementioned literature would also be useful. 2) '52 potential evapotranspiration stations' Do you mean evaporation pans? Please clarify. 3) Irrigation abstraction from the basin could be large. How is it considered in the model? 4) 'A hydraulic conductivity value of 15m/day for a 4m deep aquifer produced the best fit.' This value seems quite large. Please comment on it.

---

## Referee Comment (RC2) · Anonymous Referee #2 · 7 Jul 2016

This study explores the consequences of climate change on the discharge of Yangtze River at the Three Gorges Dam, which corresponds to a catchment of 1,007,200 km2 (this is close to the combined area of France and Germany). The distributed processbased model Shetran is forced by projections from 35 CMIP5 GCMs downscaled using a delta change approach. The sign of the annual discharge change under future conditions is uncertain (projections vary from -29.8% to 16.0% depending on the GCM). The authors attribute this uncertainty mainly to differences in how the summer monsoon is simulated by the different GCMs. Overall the study is interesting and thorough. The paper is clearly structured and well written. Yet, although the authors use a distributed model over a very large catchment, they principally discuss changes in discharge averaged over the whole catchment. I think that the hydrological processes leading to changes in discharge should be better discussed (in particular changes in

ET and in snow accumulation/melt) and that changes in subbasins could be explored too. This could provide valuable insights into into the realism of the simulations and into the sources of uncertainty affecting the projections. This study is an interesting and relevant contribution, but the processes leading to changes in discharge and to the uncertainty in the projections need to be better discussed.

**Major comments**

1. Evapotranspiration. PET is estimated using Thornthwaite equation, which is an empirical, temperature-based formulation. It is has been shown that this formulation can lead to an overestimation of PET under climate change, indicating that it reacts too strongly to temperature increase (Sheffield et al., 2012). I understand the argument developed by the authors in Section 4.2 that data is missing to use a more processbased method, but the limitations of the approach should be more clearly stated and the implications should be better explored. This is particularly important since the projected changes in PET seem to significantly influence the range of the projected future discharge. The authors report that this range changes from [-29.8%,+16%] when Thornthwaite equation is used to [-7.6%, +28.7%] when PET is held constant (page 10, lines 9-13). I suggest that: 1) the author examine whether there is any trend in the observations from the 52 PET stations (ideally over a period longer than 10 years) and 2) the authors extract the ET simulated by the climate models, and assess whether there is a increase in ET, as those simulations can be considered as more reliable than Thornthwaite approximations (Milly and Dunne, 2016). Maybe add a plot to Figure 6 showing ET as simulated by the GCMs and as simulated by Shetran, and add boxplot to Figure 8 showing future ET if PET is held constant. This would illustrate the sensitivity of the projections to the formulation of PET.

2. *Snow.* I understand from Figure 2 that winters are relatively dry in the region, hence that the influence of snow might be smaller than what would be expected from the elevation, but changes in snow accumulation and snow melt should be discussed. The authors report that "the modelling suggests that under the present climate 4.2mm of

June discharge is from snowmelt; this reduces to 2.2mm for the median of the CMIP5 simulations" (page 8, lines 12-13), which is a valuable piece of information. But typically, if more precipitation falls as rain and less as snow, this translates into an increase of winter discharge, which is hard to see in Figure 9. Could the authors comment on this? More generally, it suggest that the authors plot the annual cycle of SWE (monthly means) under current and future climate, and discuss to which extent the snow pack influences current and future discharge.

3. *GCM selection*. The authors find that the simulations of several variables by the GCMs are "implausible" (Tables 3 and 4), yet they still decide to include those models in the ensemble. I suggest conducting a second evaluation, in which they exclude the climate models that they do not deem realistic. Does it lead to a significantly lower spread of the ensemble of discharge projections? In Figure 10, implausible models fall close to the regression line, but I do not consider this as a proof of realism, since they could well fall there for the wrong reasons.

4. *Sources of uncertainty.* "So it could be argued that under a future climate the uncertainties in discharge from using Shetran are smaller than the uncertainties in the projected future climate." (page 10, line 30). This is a quite general statement, slightly speculative. Previous studies have shown that it tends to be true (e.g. Vano et al., 2014), but that in catchments with a complex topography, where snow plays a key role, the hydrological model can be a major source of uncertainty (e.g. Addor et al., 2014). It is hard to really discuss the relative importance of the differences sources of uncertainty when only one emission scenario, one downscaling method and one hydrological model are used. That said, I recognize that modeling such a large catchment under climate change in an area where comparatively little data are available is already a significant achievement, and I congratulate the authors for this. I encourage them to better explain why they decided to only sample the uncertainty stemming from the GCMs, and in particular why they decided to run a distributed process-based model, when several semi-distributed more conceptual models could probably have been run

СЗ

for a comparable computing cost.

5. *Distributed modeling.* I find it surprising that the authors chose to run a distributed model, but then barely discuss regional differences within the catchment. Given the size of the catchment and its elevation range, there are probably some interesting spatial patterns. For instance, which regions show the largest changes in terms of ET? And how much is the snow line rising as a result of higher temperature?

**Minor comments**

There is a relatively strong emphasis on floods in the text (e.g. in first sentence of the abstract and of the conclusions) but floods are not simulated nor discussed in a quantitative way, and as the authors recognize, the delta change approach is not adequate for modeling extremes (page 9, line 21). I suggest that the authors rethink the way they discuss floods.

Page 3, section 2.1: Thiessen polygons were used to account for spatial variations of precipitation and temperature within the catchment. Is it correct the forcing was considered uniform within each polygon (i.e. that no correction was applied to account for elevation changes within each polygon)? For instance, for the polygon located in the north-western corner of catchment, which has an era of about 250km x 250km, was the model, which is run on 10km grid, fed with a uniform forcing based on the measurements of a single station? If this is correct, please discuss the implications for snow modeling.

Page 3, lines Page 4 line 15: If HRUs were used, please explained how they were constructed. If not, please explain why.

Page 5, lines 4-6: "The calibration was for 1996-2000 and the validation period for 2001-2005. The comparison between measured and simulated discharge is made using the Nash Sutcliffe Efficiency (NSE)". Was any algorithm used for the calibration or was it a manual calibration?

Page 5, lines 21-24: "We analysed changes in precipitation and air temperature between 1981-2010 and 2041-2070 from 21 GCM grid cells over the Yangtze for each of the CMIP5 runs, extracted monthly change factors (ratio for precipitation, absolute for temperature) and modified the observed time series data using the monthly CF from the nearest CMIP5 grid cell." Maybe clarify whether the "observed time series" are measurements from the 64 precipitation stations and 90 air temperature stations.

Page 6, line 20: "The colouring indicates the quality of the model against observations using the same system as McSweeney et al. (2015)". Please briefly explain how the different categories were defined. In particular, explain how the colors for the second column of Table 3 (summer monsoon) were obtained.

Page 6, line 24. "It can be seen that many of the models are poor in their simulation of the monsoon". What is "poor"? Is it with "Significant biases" or "Implausible"? Please be more specific. Page 6, line 26: "all CMIP5 model runs overestimate annual observed precipitation", indeed the overestimation is quite clear and generalized across the GCMs (Figure 5a). Is it this overestimation reported by other studies focusing on the same region? Can the authors discuss its possible origins?

Page 10, line 25: "There are still uncertainties in using Shetran to predict discharges for precipitation outside the limits of the model calibration and validation period. However, as Shetran is a physically-based model, theoretically this means that the predicted discharges will be representative of future climates." I disagree with this second sentence. For instance, if PET estimates are biased, the modeled ET will most likely be biased too, and so will the simulated discharge. Also, accounting for land cover is indeed a step towards process-based modeling, but if the land cover is assumed constant under a changing climate although it might well change, this partially defeats the purpose of accounting for land cover. I think the second sentence should be removed.

Section 4.4: "The key to predicting future changes to discharge in the Yangtze basin is correctly predicting how the strength and location of the summer monsoon will change

under a future climate." The authors should consider adding that another key challenge is to better estimate future ET.

Table 2: Please indicate the parameter ranges used for the calibration.

Tables 3 and 4: Overall, I find that tables of numbers, like Tables 3 and 4 are difficult to interpret. I suggest replacing them by a graphical representation of the same content. Or at least producing a Figure similar to Figure 5 but for temperature.

Figure 1: Please add a color bar showing elevation.

Figure 5: Why are some models represented by a colored line and others by a gray line? I am guessing from the caption of Figure 10 that grey models do not have lateral boundary conditions available. Please amend the caption. Figure 5c: Why did the authors decide to depict the monthly fraction and not the monthly amounts? Without the monthly amounts it is hard to tell how well the GCMs are doing in absolute terms.

Figure 6: mm/month instead of mm?

Figure 7: Would it be possible to replace this Figure by a map, with for instance the color of the grid cells indicating the mean change, and the hatching density indicating the agreement between the different models? Or at least add some kind of information on the location of these grid points, for instance "south-west", etc.

Figure 8: The second sentence of the caption should probably be "The blue squares show the values for the present climate", like in the text. But then, which of these values are measured and which are modeled?

Figure 11: I find this comparison really interesting, but the discussion would be easier to follow if Figure 11c was replaced by a map showing the differences between the models. As already stated, I am not convinced by the choice of showing the monthly precipitation fraction instead of the monthly means (Figure 11b.)

**References**

Addor, N., Rössler, O., Köplin, N., Huss, M., Weingartner, R. and Seibert, J.: Robust changes and sources of uncertainty in the projected hydrological regimes of Swiss catchments, Water Resour. Res., 50, 7541–7562, doi:10.1002/2014WR015549, 2014.

Milly, P. C. D. and Dunne, K. A.: Potential evapotranspiration and continental drying, Nat. Clim. Chang., doi:10.1038/NCLIMATE3046, 2016.

Sheffield, J., Wood, E. F. and Roderick, M. L.: Little change in global drought over the past 60 years, Nature, 491(7424), 435–8, doi:10.1038/nature11575, 2012.

Vano, J. A., Udall, B., Cayan, D. R., Overpeck, J. T., Brekke, L. D., Das, T., Hartmann, H. C., Hidalgo, H. G., Hoerling, M., McCabe, G. J., Morino, K., Webb, R. S., Werner, K. and Lettenmaier, D. P.: Understanding uncertainties in future Colorado River streamflow, Bull. Am. Meteorol. Soc., 95(1), 59–78, doi:10.1175/BAMS-D-12-00228.1, 2014.

---

## Author Comment (AC1) · 28 Aug 2016

We would like to thank the referee for their insightful comments which we address below and which will result in an improved paper.

Main comment. "My main concern is on the short study period for the river discharge (only ten years). For climate studies, a ten year period cannot show clear patterns of climate change and environmental change (land use/land cover). Yangtze River is a major river in China and it is a surprise that the authors couldn't find more flow records beyond the ten year study period"

We agree that it not ideal for the simulated study period to consist of only ten years of measured discharge data. The three Chinese co-authors have tried to obtain longer datasets but have been unsuccessful.

The problem is discussed in detail in Section 4.3. Within this section we considered some of the problems of using only 10 years of data by comparing the 10 years of precipitation used in this work against the 30 years of data from the GPCC dataset. Both show similar inter-annual variability. In the 30 year GPCC record there are no extremes of precipitation which are large outliers to the 10 year record.

Regarding changes in land use, for the revised paper we will have a look at global land cover maps to see if there have been any significant changes.

Comment 1 "More justifications/exploration on the Shetran model would be useful. As the authors have said Other hydrological models have previously been applied to the Yangtze basin (Hayashi et al. 2008, Woo et al. 2009, Xu et al. 2008), but in terms of grid resolution, this is the most detailed hydrological model that has been produced for a major part of this basin.' It would be useful if the authors could try different model resolutions to show the resolution effect on the modelling result at this basin so that a high resolution model is justified. A comparison with other models from the aforementioned literature would also be useful"

For the revised paper we will run a coarser resolution of the Shetran model to test the effect that this will have on the results. A comparison with the result of simulations from the other models of the Yangtze will also be made.

Comment 2. "52 potential evapotranspiration stations' Do you mean evaporation pans? Please Clarify".

We will clarify this for the revised paper.

Comment 3. "Irrigation abstraction from the basin could be large. How is it considered in the model?"

Irrigation abstraction is not considered in the model. This is because the main crop growing area is in the Sichuan basin and within this part of the catchment the main growing season is during the warm wet season when water availability is not an issue

and so abstractions are not significant.

Comment 4. " 'A hydraulic conductivity value of 15m/day for a 4m deep aquifer produced the best fit.' This value seems quite large. Please comment on it. "

The problem of the lack of information about aquifers within the catchment is mentioned in the text. The calibrated value of 15m/day for a 4m aquifer gives a transmissivity of 60 m2/day which is a fairly typical value for an unconfined sandstone aquifer –which seems to be the main aquifer type.

Steve Birkinshaw (on behalf of the co-authors)

---

## Author Comment (AC2) · 28 Aug 2016

We would like to thank the referee for their comprehensive review and comments which we address below. The referee makes some excellent suggestions for improvements to the paper and we will incorporate these suggestions.

Major comment 1. "Evapotranspiration. ....I suggest that: 1) the author examine whether there is any trend in the observations from the 52 PET stations (ideally over a period longer than 10 years) and 2) the authors extract the ET simulated by the climate models, and assess whether there is a increase in ET, as those simulations can be considered as more reliable than Thornthwaite approximations (Milly and Dunne, 2016). Maybe add a plot to Figure 6 showing ET as simulated by the GCMs and as simulated by Shetran, and add boxplot to Figure 8 showing future ET if PET is held
constant. This would illustrate the sensitivity of the projections to the formulation of  $\ensuremath{\mathsf{PET}}\xspace"$

We will look at 1) Trends in the measured evapotranspiration data. 2) Trends in ET extracted from the climate models. Adding a plot to Figure 6 and a boxplot to Figure 8 are excellent suggestions which we will incorporate into the updated paper.

Major comment 2. "Snow. ... More generally, it suggest that the authors plot the annual cycle of SWE (monthly means) under current and future climate, and discuss to which extent the snow pack influences current and future discharge."

We will consider snow in more detail in the updated paper. As the referee suggests we plan to add an extra figure to show the annual cycles of SWE under current and future climates and discuss the effects of this on discharge.

Major comment 3 "GCM selection. The authors find that the simulations of several variables by the GCMs are "implausible" (Tables 3 and 4), yet they still decide to include those models in the ensemble. I suggest conducting a second evaluation, in which they exclude the climate models that they do not deem realistic. Does it lead to a significantly lower spread of the ensemble of discharge projections? In Figure 10, implausible models fall close to the regression line, but I do not consider this as a proof of realism, since they could well fall there for the wrong reasons."

We will consider another evaluation of the effect of the GCMs on discharge in which we exclude the climate models that are not considered to be realistic. We agree that in Figure 10 if implausible GCMs fall close to the regression line it is not considered as a proof of realism.

Major comment 4 "Sources of uncertainty. ... I encourage them to better explain why they decided to only sample the uncertainty stemming from the GCMs, and in particular why they decided to run a distributed process-based model, when several semidistributed more conceptual models could probably have been run"
In the updated paper we will modify the text to explain our choice of using a single model with uncertainty only stemming from the GCMs.

Major comment 5 "Distributed modeling. I find it surprising that the authors chose to run a distributed model, but then barely discuss regional differences within the catchment. Given the size of the catchment and its elevation range, there are probably some interesting spatial patterns. For instance, which regions show the largest changes in terms of ET? And how much is the snow line rising as a result of higher temperature?"

The regional differences are considered in Figure 7 and the corresponding text, where the numbers on the x axis refer to locations within the catchment. We will update the paper to discuss this spatial variation in more detail. As discussed in Major comment 2 an additional Figure showing spatial variations in snow will be added.

Minor comments 1. "There is a relatively strong emphasis on floods in the text (e.g. in first sentence of the abstract and of the conclusions) but floods are not simulated nor discussed in a quantitative way, and adequate for modeling extremes (page 9, line 21). I suggest that the authors rethink the way they discuss floods"

We will change the focus in the text to remove the emphasis away from flooding.

Minor comments 2. "Page 3, section 2.1: Thiessen polygons were used to account for spatial variations of precipitation and temperature within the catchment. Is it correct the forcing was considered uniform within each polygon (i.e. that no correction was applied to account for elevation changes within each polygon)? For instance, for the polygon located in the north-western corner of catchment, which has an era of about 250km x 250km, was the model, which is run on 10km grid, fed with a uniform forcing based on themeasurements of a single station? If this is correct, please discuss the implications for snow modeling."

This is correct. We will discuss the significance in terms of snow modelling

Minor comments 3. "Page 4 line 15: If HRUs were used, please explained how they
were constructed. If not, please explain why."

We are not sure if we fully understand this comment. Shetran is a spatial distributed model and each 10km by 10km grid square has its own individual land use and soil/aquifer type.

Minor comments 4. "Page 5, lines 4-6: "The calibration was for 1996-2000 and the validation period for 2001-2005. The comparison between measured and simulated discharge is made using the Nash Sutcliffe Efficiency (NSE)". Was any algorithm used for the calibration or was it a manual calibration?"

The text will be modified to make it clear than it was a manual calibration

Minor comments 5. "Page 5, lines 21-24: "We analysed changes in precipitation and air temperature be-tween 1981-2010 and 2041-2070 from 21 GCM grid cells over the Yangtze for each of the CMIP5 runs, extracted monthly change factors (ratio for precipitation, absolute for temperature) and modified the observed time series data using the monthly CF from the nearest CMIP5 grid cell." Maybe clarify whether the "observed time series" are measurements from the 64 precipitation stations and 90 air temperature stations"

We will clarify that the measurements are from the 64 precipitation stations and 90 air temperature stations

Minor comments 6. "Page 6, line 20: "The colouring indicates the quality of the model against observations using the same system as McSweeney et al. (2015)". Please briefly explain how the different categories were defined. In particular, explain how the colors for the second column of Table 3 (summer monsoon) were obtained"

We are planning to reconsider the colours in the 3rd, 4th and 6th columns in Table 3. We agree that currently they are not well defined.

Minor comments 7. "Page 6, line 24. "It can be seen that many of the models are poor in their simulation of the monsoon". What is "poor"? Is it with "Significant biases" or
"Implausible"? Please be more specific. Page 6, line 26: "all CMIP5 model runs overestimate annual observed precipitation", indeed the overestimation is quite clear and generalized across the GCMs (Figure 5a). Is it this overestimation reported by other studies focusing on the same region? Can the authors discuss its possible origins?"

We will be more specific about the explaining the term "poor" in the revised paper. We will also try to find more details on the origins of the overestimation of annual observed precipitation.

Minor comments 8. "Page 10, line 25: "There are still uncertainties in using Shetran to predict discharges for precipitation outside the limits of the model calibration and validation period. However, as Shetran is a physically-based model, theoretically this means that the predicted discharges will be representative of future climates." I disagree with this second sentence. For instance, if PET estimates are biased, the modeled ET will most likely be biased too, and so will the simulated discharge. Also, accounting for land cover is indeed a step towards process-based modeling, but if the land cover is assumed constant under a changing climate although it might well change, this partially defeats the purpose of accounting for land cover. I think the second sentence should be removed"

We will remove the second sentence.

Minor comments 9. "Section 4.4: "The key to predicting future changes to discharge in the Yangtze basin is correctly predicting how the strength and location of the summer monsoon will change under a future climate." The authors should consider adding that another key challenge is to better estimate future ET"

In the revised paper we will add that the better estimation of future ET is another key challenge.

Minor comments 10. "Table 2: Please indicate the parameter ranges used for the calibration"
In the revised paper we will make this change.

Minor comments 11. "Tables 3 and 4: Overall, I find that tables of numbers, like Tables 3 and 4 are difficult to interpret. I suggest replacing them by a graphical representation of the same content. Or at least producing a Figure similar to Figure 5 but for temperature"

We plan to add a figure similar to Figure 5 but for temperature.

Minor comments 12. "Figure 1: Please add a color bar showing elevation"

We will add a colour bar showing elevation.

Minor comments 13. "Figure 5: Why are some models represented by a colored line and others by a gray line? I am guessing from the caption of Figure 10 that grey models do not have lateral boundary conditions available. Please amend the caption. Figure 5c: Why did the authors decide to depict the monthly fraction and not the monthly amounts? Without the monthly amounts it is hard to tell how well the GCMs are doing in absolute term"

If all the GCMs were coloured it is impossible to distinguish which line is associated with which model. So only the best and the worst models are coloured. We will amend the caption to make this clear.

We originally showed the monthly amounts not the monthly fractions but the monthly response was overwhelmed by the annual totals (which are already shown in Figure 5a). For example in a GCM model where the annual totals were twice the measured total all that could be seen were the monthly totals were larger than the measured totals.

Minor comments 14. "Figure 6: mm/month instead of mm?"

We will change the axis title

Minor comments 15. "Figure 7: Would it be possible to replace this Figure by a map, with for instance the color of the grid cells indicating the mean change, and the hatching
density indicating the agreement between the different models? Or at least add some kind of information on the location of these grid points, for instance "south-west", etc"

We agree that currently it is not clear where the numbers on the x axis refer to. We will look at ways of changing the Figure so it is easier to interpret.

Minor comments 16." Figure 8: The second sentence of the caption should probably be "The blue squares show the values for the present climate", like in the text. But then, which of these values are measured and which are modeled?"

We will change the caption to make this correct and clear

Minor comments 17. "Figure 11: I find this comparison really interesting, but the discussion would be easier to follow if Figure 11c was replaced by a map showing the differences between the models. As already stated, I am not convinced by the choice of showing the monthly precipitation fraction instead of the monthly means (Figure 11b)."

We agree that it would be clearer by showing the data in Figure 11c on a map. We will think of the best way of doing this.

Steve Birkinshaw (on behalf of the co-authors)

HESSD

---

## Editor Comment (EC1) · J. Seibert (Editor) · 4 Sep 2016

Both reviewers provide detailed and valuable comments. The author's responses indicate that the author's will be able to use these comments to improve the manuscript. An important point are some assumptions, which might questionable. Here a frank discussion including the implications for the results is needed.

---

## Author Response (AR1)

**This document is the response of the authors to the reviewers comments. It includes a point-by-point response to the reviews and a marked-up manuscript version**

**Referee 1**

We would like to thank the referee for their insightful comments which we address below and which have resulted in an improved paper.

**Main comment:** *"My main concern is on the short study period for the river discharge (only ten years). For climate studies, a ten year period cannot show clear patterns of climate change and environmental change (land use/land cover). Yangtze River is a major river in China and it is a surprise that the authors couldn't find more flow records beyond the ten year study period"*

**Authors Response:** We agree that it not ideal for the simulated study period to consist of only ten years of measured discharge data. The three Chinese co-authors have tried to obtain longer datasets but have been unsuccessful. The problem is discussed in detail in Section 4.3. Within this section we considered some of the problems of using only 10 years of data by comparing the 10 years of precipitation used in this work against the 30 years of data from the GPCC dataset. Both show similar inter-annual variability. In the 30 year GPCC record there are no extremes of precipitation which are large outliers to the 10 year record.

The analysis of global land cover maps show that there have only been small changes in the land-use in recent decades. The following has been added to the text together with the corresponding references "There has been little change in land-use within the Yangtze basin in recent decades (Hansen et al. 2013). The most significant change has been the urbanization within the Sichuan basin but this increase covers less than 1% of the Sichuan basin (Liu et al. 2010)."

**Comment 1:** *"More justifications/exploration on the Shetran model would be useful. As the authors have said Other hydrological models have previously been applied to the Yangtze basin (Hayashi et al. 2008, Woo et al. 2009, Xu et al. 2008), but in terms of grid resolution, this is the most detailed hydrological model that has been produced for a major part of this basin.' It would be useful if the authors could try different model resolutions to show the resolution effect on the modelling result at this basin so that a high resolution model is justified. A comparison with other models from the aforementioned literature would also be useful"*

**Authors Response:** The Shetran simulations in the paper use a 10km x 10km grid. We have carried simulation using Shetran with a 20km x 20km grid and a 40km by 40km grid. These models are significantly less good. A comparison with other models was also carried out. The following has been added to the text "As well as using a 10km by 10km grid, Shetran simulations were carried out using both a 20kmx 20km grid and a 40km x 40km grid. The results for coarser grid resolutions were less good, with an overall (1996-2005) NSEs of 0.79 and 0.66, respectively. This was mainly due to a poorer connectivity between the land surface grid squares and the river channels resulting in a much smoother hydrograph, with the simulated peak also occurring later than the measured peak. The results are also better than other models of the Yangtze basin with a coarser grid resolution. Woo et al. (2009)'s SLURP model gives a NSE of 0.83 and Xu et al. (2008)'s GBHM model a NSE of 0.85"

**Comment 2.** *"52 potential evapotranspiration stations' Do you mean evaporation pans? Please Clarify"*.

**Authors Response:** The text has been changed to clarify that evaporation pans were used.

**Comment 3.** *"Irrigation abstraction from the basin could be large. How is it considered in the model?"*

**Authors Response:** Irrigation abstraction is not considered in the model. This is because the main crop growing area is in the Sichuan basin and this is currently not an important issue. No changes to the text have been made

**Comment 4.** *" 'A hydraulic conductivity value of 15m/day for a 4m deep aquifer produced the best fit.' This value seems quite large. Please comment on it. "*

**Authors Response:** The problem of the lack of information about aquifers within the catchment is mentioned in the text. The calibrated value of 15m/day for a 4m aquifer gives a transmissivity of 60 m2/day which is a fairly typical value for an unconfined sandstone aquifer –which seems to be the main aquifer type. No changes to the text have been made

Steve Birkinshaw (on behalf of the co-authors)

**Referee 2**

We would like to thank the referee for their comprehensive review and comments which we address below. The referee makes some excellent suggestions for improvements to the paper and we have incorporated these suggestions.

**Major comment 1.** *"Evapotranspiration. ....I suggest that: 1) the author examine whether there is any trend in the observations from the 52 PET stations (ideally over a period longer than 10 years) and 2) the authors extract the ET simulated by the climate models, and assess whether there is a increase in ET, as those simulations can be considered as more reliable than Thornthwaite approximations (Milly and Dunne, 2016). Maybe add a plot to Figure 6 showing ET as simulated by the GCMs and as simulated by Shetran, and add boxplot to Figure 8 showing future ET if PET is held constant. This would illustrate the sensitivity of the projections to the formulation of PET"*

**Authors Response:** There is insufficient data (only 10 years) to look at trends in the 52 PET stations. As the referee suggests we have extracted the ET values from the 78 climate model projections (35 different GCMs) for the current (1981-2010) and future climates (2041-2070). Averaged over the entire Yangtze basin these show an 8.4% increase compared to a 17% increase using the Thornthwaite equation. The following has been added to the text "Further analysis was carried out by considering the change in actual evaporation from the 78 climate model projections. Averaged over the Yangtze basin this shows an increase of 8.4% under the future climate compared to a 17% increase in actual evaporation using the PET calculated from the Thornthwaite equation and a 1% increase in actual evaporation with no change in PET. This suggests the future actual evaporation might be between the two extremes shown in Fig 10a and Fig10b." As suggested by the referee we have added another Box plot to Figure 10 (previously Figure 8) showing future ET if the PET is held constant.

**Major comment 2.** *"Snow. ... More generally, it suggest that the authors plot the annual cycle of SWE (monthly means) under current and future climate, and discuss to which extent the snow pack influences current and future discharge."*

**Authors Response:** As suggested by the referee we have added an additional Figure of the annual cycle of SWE under the current climate and for two GCMs (CanESM2 and CSIRO Mk3-6-0) under a future climate. The effect of snow under the current climate has been considered in more detail with the following added to the text in Section 3.1

"Accumulation of snow in the winter is a significant process in approximately 25% of the catchment (above around 3000m in the north of the basin and 4500m in the south of the basin) – there are occasional snow falls in other parts of the basin. This can be seen in Fig. 5a which shows the monthly accumulations at the end of the month from December to May over the 10-year simulation. On average the maximum snow water equivalent depth is 50mm at the end of March. The totals are slightly lower in the Tibetan plateau as the winter precipitation totals are lower than for the area further east. Over the entire basin the spatially averaged snow water equivalent depth is 6.6mm at the end of March compared to a spatially averaged precipitation of 29mm and an annual precipitation total of 904mm. As significant snow accumulation takes place during the dry part of the year in the drier part of the Yangtze basin, the effect of snow accumulation and melt on discharge at Yichang is less than might be expected from considering just the temperature within the basin. Within the model the simulated snow accumulation and melting depend only on the precipitation and temperatures calculated for each grid square using a Thiessen polygon approach. Within each polygon there is no simple relationship between elevation and precipitation to improve this approach. The 90 temperature stations give a good representation of the spatial distribution of temperature in the basin. Where there is a sparse data in the Tibetan plateau there is a small range of elevations"

The following text has been added in Section 3.3

"Figure 5 shows that by the end of May under the present climate there is still a significant amount of snow in the upper part of the basin whereas for CanESM2 and CSIRO-Mk3-6-0 (+12.7 and-3.7% change in precipitation, respectively) all the remaining snow melts in May (note that there is a travel time of approximately 30 days for the water to flow from the upper part of the basin to Yichang)."

**Major comment 3** *"GCM selection. The authors find that the simulations of several variables by the GCMs are "implausible" (Tables 3 and 4), yet they still decide to include those models in the ensemble. I suggest conducting a second evaluation, in which they exclude the climate models that they do not deem realistic. Does it lead to a significantly lower spread of the ensemble of discharge projections? In Figure 10, implausible models fall close to the regression line, but I do not consider this as a proof of realism, since they could well fall there for the wrong reasons."*

**Authors Response:** We have considered another evaluation of the effect of the GCMs on discharge in which we exclude the climate models that are not considered to be realistic. The response is very similar to the existing Figure and does not lead to a significantly lower spread of the ensemble of discharge projections. The sentence has been modified: "The 'satisfactory' green points cover almost the entire range and so it is very hard to discount any future projections of change in precipitation or discharge (plotting only these points gives a very similar response and does not lead to a significantly lower spread of the ensemble of discharge projections)."

We agree that in Figure 10 if implausible GCMs fall close to the regression line it is not considered as a proof of realism.

**Major comment 4** *"Sources of uncertainty. ... I encourage them to better explain why they decided to only sample the uncertainty stemming from the GCMs, and in particular why they decided to run a distributed process-based model, when several semi-distributed more conceptual models could probably have been run"*

**Authors Response:** We agree that there are many sources of uncertainly when considering the effect of future climate on discharge. The approach we took is that it is best to focus on the uncertainty stemming only from the GCMs but to do a complete and thorough analysis of this uncertainty. This is because, for this area, the uncertainty coming from the climate models projections is very high, without agreement even in the direction of change in precipitation. Also, we selected a distributed hydrological model that enables the hydrological processes within the Yangtze to be captured in the most physically realistic way and showed an excellent match between the measured and simulated discharge under the current climate. This model results were considerably better than that achieved by other authors for the same basin. So we have added the following "Woo et al. (2009)'s SLURP model gives a NSE of 0.83 and Xu et al. (2008)'s GBHM model a NSE of 0.85" whereas here a NSE of 0.92 was achieved for the validation period.

The following text considering model uncertainly also been added/modified: "One of the major sources of uncertainty when predicting future discharge as a result of climate change is model uncertainty. In this work the main uncertainties in using Shetran is predicting discharges for precipitation outside the limits of the model calibration and validation period"

**Major comment 5:** *"Distributed modeling. I find it surprising that the authors chose to run a distributed model, but then barely discuss regional differences within the catchment. Given the size of the catchment and its elevation range, there are probably some interesting spatial patterns. For instance, which regions show the largest changes in terms of ET? And how much is the snow line rising as a result of higher temperature?"*

**Authors Response:** The regional differences are considered in Figure 7 and the corresponding text, where the numbers on the x axis refer to locations within the catchment. As discussed in Major comment 2 an additional Figure showing spatial variations in snow has be added and the text changed to highlight the regional differences.

**Minor comments 1.** *" There is a relatively strong emphasis on floods in the text (e.g. in first sentence of the abstract and of the conclusions) but floods are not simulated nor discussed in a quantitative way, and adequate for modeling extremes (page 9, line 21). I suggest that the authors rethink the way they discuss floods"*

**Authors Comment:** We have removed flooding from the text in the abstract.

**Minor comments 2.** *"Page 3, section 2.1: Thiessen polygons were used to account for spatial variations of precipitation and temperature within the catchment. Is it correct the forcing was considered uniform within each polygon (i.e. that no correction was applied to account for elevation changes within each polygon)? For instance, for the polygon located in the north-western corner of catchment, which has an era of about 250km x 250km, was the model, which is run on 10km grid, fed with a uniform forcing based on the measurements of a single station? If this is correct, please discuss the implications for snow modeling."*

**Authors Response:** Thiessen polygons were used to account for spatial variations of precipitation and temperature within the catchment. In terms of precipitation in the Yangtze basin there is no increase with an increased elevation. The location within the catchment rather than the elevation is important so a simple Theissen polygon approach is the best approach. In terms of temperature 90 stations are used which overall gives a good representation of temperature in the basin. Where there is sparse data in the Tibetan plateau there is a small range of elevations, so any temperature correction would be small (and so only small changes in the timing of the snow melt would be expected). The changes to the text are already highlighted in Major comment 2.

**Minor comments 3.** *"Page 4 line 15: If HRUs were used, please explained how they were constructed. If not, please explain why."*

**Authors Response:** We are not sure if we fully understand this comment. Shetran is a spatial distributed model and each 10km by 10km grid square has its own individual land use and soil/aquifer type.

**Minor comments 4.** *"Page 5, lines 4-6: "The calibration was for 1996-2000 and the validation period for 2001-2005. The comparison between measured and simulated discharge is made using the Nash Sutcliffe Efficiency (NSE)". Was any algorithm used for the calibration or was it a manual calibration?"*

**Authors Response:** The text has been modified with "manual calibration" added

**Minor comments 5.** *"Page 5, lines 21-24: "We analysed changes in precipitation and air temperature be-tween 1981-2010 and 2041-2070 from 21 GCM grid cells over the Yangtze for each of the CMIP5 runs, extracted monthly change factors (ratio for precipitation, absolute for temperature) and modified the observed time series data using the monthly CF from the nearest CMIP5 grid cell." Maybe clarify whether the "observed time series" are measurements from the 64 precipitation stations and 90 air temperature stations"*

**Authors Response:** The following text has been added: "64 precipitation stations and 90 temperature stations"

**Minor comments 6.** *"Page 6, line 20: "The colouring indicates the quality of the model against observations using the same system as McSweeney et al. (2015)". Please briefly explain how the different categories were defined. In particular, explain how the colors for the second column of Table 3 (summer monsoon) were obtained"*

**Authors Response:** We have removed most of the columns in Table 3 and 4, we agree that they were not well defined. The following text has been added/modified to explain how the different categories were defined and in particular how the colours for the second column of Table 3 were obtained . "In the second column of Table 3 we reproduce results from McSweeney et al. (2015) to indicate the performance of the GCMs at reproducing large scale circulation flow at 850 hPa for the Asian summer monsoon. This flow is largely westerly across peninsular India before diverting to a south-westerly flow across the Bay of Bengal, and then to westerly across continental south-east Asia. The colours are 'Satisfactory' (green),

'Biases' (yellow), 'Significant biases' (orange) and 'Implausible' (red) and the grey colour means the model was not available to McSweeney et al. (2015). The 'Implausible' models have an unrealistic representation of the large-scale flows of the monsoon and those with 'biases' are not able to reproduce the strength of flows."

5 **Minor comments 7.** *"Page 6, line 24. "It can be seen that many of the models are poor in their simulation of the monsoon". What is "poor"? Is it with "Significant biases" or "Implausible"? Please be more specific. Page 6, line 26: "all CMIP5 model runs overestimate annual observed precipitation", indeed the overestimation is quite clear and generalized across the GCMs (Figure 5a). Is it this overestimation reported by other studies focusing on the same region? Can the authors discuss its possible origins?"*

10 **Authors Response:** This sentence has been removed: "It can be seen that many of the models are poor in their simulation of the monsoon" and replaced by the sentences in Minor comment 6

Regarding the overestimation of annual observed precipitation by GCMs the issue is complex and not clearly understood and we feel it is out of the scope of this paper.

15 **Minor comments 8.** *"Page 10, line 25: "There are still uncertainties in using Shetran to predict discharges for precipitation outside the limits of the model calibration and validation period. However, as Shetran is a physically-based model, theoretically this means that the predicted discharges will be representative of future climates." I disagree with this second sentence. For instance, if PET estimates are biased, the modeled ET will most likely be biased too, and so will the simulated discharge. Also, accounting for land cover is indeed a step towards process-based modeling, but if the land cover*
20 *is assumed constant under a changing climate although it might well change, this partially defeats the purpose of accounting for land cover. I think the second sentence should be removed"*

**Authors Response:** As suggested by the referee we have removed the second sentence.

**Minor comments 9.** *"Section 4.4: "The key to predicting future changes to discharge in the Yangtze basin is correctly*
25 *predicting how the strength and location of the summer monsoon will change under a future climate." The authors should consider adding that another key challenge is to better estimate future ET"*

**Authors Response:** Sections 4.2 and 4.4 have been modified to make it clear that another key challenge is to better estimate future PET.

30 **Minor comments 10.** *"Table 2: Please indicate the parameter ranges used for the calibration"*

**Authors Response:** As suggested by the referee we have made this change.

**Minor comments  11.** *"Tables 3 and 4: Overall, I find that tables of numbers, like Tables 3 and 4 are difficult to interpret. I suggest replacing them by a graphical representation of the same content. Or at least producing a Figure similar to Figure 5 but for temperature"*

**Authors Response:** As suggested we have added a figure similar to Figure 5 but for temperature. (Figure 5 is now called Figure 6 and there is a new Figure 7)

**Minor comments  12.** *"Figure 1: Please add a color bar showing elevation"*

**Authors Response:** As suggested we have added a colour bar showing elevation.

**Minor comments  13.** *"Figure 5: Why are some models represented by a colored line and others by a gray line? I am guessing from the caption of Figure 10 that grey models do not have lateral boundary conditions available.  Please amend the caption.  Figure 5c:  Why did the authors decide to depict the monthly fraction and not the monthly amounts?  Without the monthly amounts it is hard to tell how well the GCMs are doing in absolute term"*

**Authors Response:** If all the GCMs in Figure 5 (now Figure 6) were coloured it is impossible to distinguish which line is associated with which model. So only the best and the worst models are coloured (this is now done in a consistent way). We have amended the caption to make this clear.

We could not decide if monthly totals or monthly fractions was the better way of showing this data. As suggested by the referee we have changed this to monthly totals.

**Minor comments  14.** *"Figure 6: mm/month instead of mm?"*

**Authors Response:** As suggested by the referee we have changed the title.

**Minor comments  15.** *"Figure 7:  Would it be possible to replace this Figure by a map, with for instance the color of the grid cells indicating the mean change, and the hatching density indicating the agreement between the different models? Or at least add some kind of information on the location of these grid points, for instance "south-west", etc"*

**Authors Response:** We have added an inset map to the figure showing where in the Yangtze basin the numbers on the x-axis refer to. We feel this makes it much easier to interpret. The reason we did not substitute the boxplots for a map was because we wanted the full range of projected changes to be visible.

**Minor comments  16.** *"Figure 8:  The second sentence of the caption should probably be "The blue squares show the values for the present climate", like in the text. But then, which of these values are measured and which are modeled?"*

**Authors Response:** As suggested by the referee we have changed the text in the Figure so it says "The blue squares show the simulated values for the present climate."

**Minor comments  17.** *"Figure 11: I find this comparison really interesting, but the discussion would be easier to follow if Figure 11c was replaced by a map showing the differences between the models.  As already stated, I am not convinced by the choice of showing the monthly precipitation fraction instead of the monthly means (Figure 11b)."*

**Authors Response:** We agree with the referee and have changed Figure 11 (now Figure 13) to include parts e) and f) so that it is much easier to see the changes in Figure 11c. As suggested by the referee monthly precipitation totals are now shown in Figure 11b (rather than monthly fractions).

[revised manuscript text omitted]

---

## Author Response (AR2)

**This document is the response of the authors to the reviewers comments on the resubmitted article. It includes a point-by-point response to the reviews and a marked-up manuscript version**

**Report 1 (Anonymous referee 2)**

We would like to thank the referee again for their very helpful comments and suggestions for improvements.

**Comment 1:** "*Line 14, page 3: "Within each polygon there is no simple relationship between elevation and precipitation to improve this approach." This sentence is unclear. I would simply state that precipitation and temperature are considered to be uniform within each polygon.*"

**Authors Response:** As suggested by the referee we have made this change. The change is on Page 6 Line 31.

**Comment 2:** "*Caption of Figure 7: Since this Figure shows temperature simulations, its caption should be "largest and smallest temperature bias" instead of "largest and smallest totals.*"

**Authors Response:** We have made this change

**Comment 3:** "*Lines 5-7, page 20: "One of the major sources of uncertainty when predicting future discharge as a result of climate change is model uncertainty. In this work the main uncertainties in using Shetran is predicting discharges for precipitation outside the limits of the model calibration and validation period". This sentence is confusing, because the source of uncertainty that this study is focusing on is the uncertainty stemming from the climate models, and uncertainties in the hydrological model (Shertan) are not explicitly considered (the authors only use one model and one parameter set). I would suggest to rephrase this sentence as follows: "Discharge simulations under future climate are uncertain, because of uncertainties in future greenhouse gas emissions, climate models, downscaling methods and hydrological models. This study focuses on the uncertainty stemming from the climate models because of its significant influence on the uncertainty in discharge projections (e.g. Ragettli et al., (2013); Addor et al., (2014))*"

**Authors Response:** As suggested by the referee we have made this change, including adding both references. The change is on Page 11 Lines 20-24.

**Comment 4:** "*Lines 23-24, page 21: "An improvement in climate model performance would also allow PET to be calculated using the Penman-Monteith equation which would deliver more reliable projections of change in PET". Penman-Montheith could already be used with the current set of climate projections. I would simply state that using a more-process based formulation of PET (e.g. Penman-Monteith) would improve the realism of the discharge projections.."*

**Authors Response:** We have made this change which is on Page 13 Line 10

**Comment 5:** "*Figure 10 has now three panels, the two bottom ones (a and b) are described in the caption but the not the top one. Please modify the figure and/or the caption.."*

**Authors Response:** I am confused by this as there only seem to be two panels in Figure 10. So no changes have been made.

Steve Birkinshaw (on behalf of the co-authors)

**Report 2 (Anonymous referee 3)**

We would like to thank the referee for their very helpful comments and suggestions for improvements.

5 **Comment 1:** "*First, I am not very familiar with the distributed model the authors used. My understanding is that although it is a distributed model, the parameters calibrated in Table 2 had the same values for all the grid cells. Is this correct? Also, I am not comfortable with the values for other parameters used by the authors, meaning that physical processes might not be correctly represented in the model. For instance, in Table 1, LAI for evergreen forest is from 0.5 to 1.0 and canopy storage capacity is 3 mm. Those values are more like for grass!*

10 *For actual/potential ET ratio, the value of 1 seems unrealistic, even for the humid region like the Yangtze River basin. I would encourage the authors to provide a supplemental table to show the details on the temporal dynamics of those parameters.*"

**Authors Response:** The referee is correct that the parameters in Table 2 are the same for all grid cells. Although for those grid squares where there is no subsurface soil there is assumed to be no aquifer present (Sect 2.2). The

15 caption in Table 2 has been modified to make this clear.

The LAI is usually defined as the leaf area per unit ground surface area. In Shetran there are two parameters which make up the LAI, these are the fraction of the ground surface covered by the vegetation  (called PLAI in Shetran)and the total leaf area to area of ground covered by vegetation (called CLAI in Shetran). We agree that the leaf area index heading in Table 1 was incorrect. It has been changed to the "Ground coverage fraction". The

20 actual LAI used in the model is five times this fraction which is realistic for an evergreen forest.

The water balance data (Figure 2) shows that during the wet season that actual evaporation (precipitation-discharge) is similar to the potential evaporation. So an actual/potential ET ratio of 1 for forest seems realistic. In Table 2 it also states that this is the value at field capacity and it reduces as the soil dries so a lower value will automatically be used during the dry season.

25 Although the temporal dynamics of the vegetation parameters play a part in the hydrological modelling we feel it is not the main focus of the paper and so we have not added an additional table.

**Comment 2.** "*Secondly, the authors claimed that the land use change does not have important impacts on the flow discharges in the Yangtze River. However, there are numerous studies that beg the difference in the same region. More importantly, the reference of Hansen et al. (2013) as cited by the authors is not appropriate here, which only shows changes in forest. Instead, as also noticed by the authors, Liu et al. (2010) shows considerable*

5  *land use changes in the study region. In terms of the orographic effect on precipitation, I am not conceived by the authors' argument. I would suggest, at least, the authors to plot elevation vs. precipitation in the paper to support their claim.*"

**Authors Response:** We have changed the text to make it clear that the Hansen et al. (2013) reference only considers forest.

10  As suggested by the Referee we have added to Figure 2 a plot showing the elevation vs. the precipitation and we have also changed the text (Page 3 Line 15) so it now says the following "So there is a general trend of decreasing annual precipitation with increasing elevation but there is also considerable variation depending on the location of the precipitation station within the basin (Figure 2b)"

15  **Comment 3** *"Lastly, the authors attributed the differences in projected discharge to the differences in projected spatial distributions of future precipitation from different GCMs; however, little explanations behind the physical mechanism is offered by the authors. Given it is the key finding of this study, I would invite the authors to do so in the next revision."*

**Authors Response:** We agree that the differences in projected spatial distributions of future precipitation from

20  different GCMs is a key finding of this study. But the issue is complex and not clearly understood and we feel it is out of the scope of this paper.

Steve Birkinshaw (on behalf of the co-authors)

[revised manuscript text omitted]

---

## Author Response (AR3)

**This document is the response of the authors to the editor's comments**

We would like to thank the Prof. Jan Seibert for his comments.

**Comment 1:** "The good performance of SHETRAN is interesting and also the decrease of performance when going to a coarser resolution. Given that even the finer resolution is rather coarse from a process point of view, the reasons for this are not obvious to me. Overall, there are some questions and possible tests regarding the model performance, which I would find interesting. However, I realize that the SHETRAN performance itself is not the focus of this study, but this could perhaps be worthwhile to explore more in another study. "

**Authors Response:** We agree that this is very interesting and should be explored in another study

**Comment 2:** "Be consistent with the capitals when writing SHETRAN (or Shetran)"

**Authors Response:** Shetran is now used throughout

**Comment 3:** "There are some other places where you used capitals and I would rather not (e.g., River in the beginning of the intro) and Snow water equivalent in the figure captions)"

**Authors Response:** These changes have been made

**Comment 4:** "It should be Thiessen (instead of Theissen)"

**Authors Response:** These errors have been corrected

**Comment 5:** "the Thiessen polygon method is a method to compute weights for the different stations, not necessarily to produce maps of spatial variations (such maps with the large jumps look a bit strange, for a map, interpolation methods would be more appropriate). However, this does not influence the results of the study."

**Authors Response:** I have discussed this point with my co-authors. We agree that the maps in Figure 3 look a bit strange and interpolation methods would be better to show the spatial variations in precipitation and temperature. However, we have decided that Thiessen polygons were used to produce the inputs for Shetran and therefore the maps should reflect that. We have changed the caption to make this clear.

Steve Birkinshaw (on behalf of the co-authors)